nature

## human behaviour
# Mobile phone data reveal the effects of violence on internal displacement in Afghanistan

Xiao Hui Tai ⬤, Shikhar Mehra and Joshua E. Blumenstock ⬤ ✉

**Nearly 50 million people globally have been internally displaced due to conflict, persecution and human rights violations. However, the study of internally displaced persons—and the design of policies to assist them—is complicated by the fact that these people are often underrepresented in surveys and official statistics. We develop an approach to measure the impact of violence on internal displacement using anonymized high-frequency mobile phone data. We use this approach to quantify the short- and long-term impacts of violence on internal displacement in Afghanistan, a country that has experienced decades of conflict. Our results highlight how displacement depends on the nature of violence. High-casualty events, and violence involving the Islamic State, cause the most displacement. Provincial capitals act as magnets for people fleeing violence in outlying areas. Our work illustrates the potential for non-traditional data sources to facilitate research and policymaking in conflict settings.**

Every year, tens of millions of individuals and families are forcibly displaced by armed conflict, creating enormous humanitarian, social and economic costs[1,2]. Recent estimates indicate that the number of people living in proximity to conflict has doubled since 2007[3]. Despite the global scale and importance of this crisis, empirical analysis of the link between violence and displacement is complicated by the inherent difficulty of observing displaced populations—a difficulty that is compounded in developing countries and in insecure environments[3,4].

This paper develops and tests an approach to studying the impact of violence on internal displacement. Our first contribution is methodological and illustrates how high-frequency 'digital trace' data can enable different approaches to identifying and estimating the causal effect of violent events on internal displacement. We use a panel event study framework that is feasible only because we observe the locations of millions of individuals near-continuously over time: the high-frequency data allow us to draw causal inferences from discontinuities in spatio-temporal trajectories that coincide with specific violent events. Such an estimation strategy would not be feasible with traditional survey-based data, which track a relatively small number of individuals at infrequent intervals. This complements recent qualitative[5,6], survey-based[7–9] and observational studies[10–12] on conflict and displacement in the developing world. It also builds on recent work using non-traditional sources of digital data to study the movement of human populations[13–24].

Our second contribution is to provide rich evidence on the impact of violence on internal displacement in Afghanistan—a country that has experienced decades of conflict and that contains over two million internally displaced people[1]. We contribute a quantitative perspective on the nature of this displacement, complementing more traditional approaches based on surveys and administrative reporting[25–27].

Our analysis is based on the universe of mobile phone activity from Afghanistan's largest mobile phone operator for a four-year period from April 2013 to March 2017. This dataset contains the anonymized mobile phone metadata of approximately ten million mobile subscribers (a non-random subset of all individuals in Afghanistan, as we discuss in greater detail below). We separately obtain geo-coded information on fatal violent events in Afghanistan,

which is collected by the Uppsala Conflict Data Program (UCDP) from public media reports[28]. We use 3,354 events in our analysis, corresponding to the subset of all events that are recorded with sufficient spatial and temporal precision (53% of all recorded events) and which overlap with our phone data; the limitations of the UCDP data are discussed in the 'Limitations' section of this paper. The spatial distribution of mobile phone towers and violent events can be seen in Fig. 1 and Supplementary Fig. 1.

The empirical approach we develop uses the mobile phone data to observe the movement of subscribers between regions, and a statistical model to estimate the causal effect of violence on this movement (see Methods for a detailed description). We first use metadata on the sequence of cell towers used by each mobile subscriber to identify each subscriber's home district (the smallest administrative unit in Afghanistan). We then adapt recent algorithmic advances in the measurement of migration from digital trace data[29] to identify days on which the subscriber's home district changes—we refer to this as a 'migration'. (The International Organization for Migration (IOM) defines migration as "the movement of persons away from their place of usual residence, either across an international border or within a State"[30]. Our main analysis focuses on migrations that involve a person leaving their origin district for at least a week; we validate our empirical measures of migration with IOM statistics in the Methods, section 'Data validation' and Supplementary Fig. 2.) These individual migration events are then aggregated to produce estimates of the total population flows between each pair of districts on each day. Finally, we use a high-frequency panel event study design to estimate the causal effect of violence on migration, which we measure as the increase in movement of subscribers out of a district impacted by violence, relative to movement from the same district on non-violent days, while controlling for seasonality and other temporal factors. Specifically, we regress total out-migration from each district on each day on a vector of binary indicators for whether violence occurred in that district on that day, in the 180 days prior (the 'lags') or the 30 days after (the 'leads') that day. We control for district and day fixed effects and cluster standard errors at the district level. This approach allows us to estimate the 'average displacement effect' of violence—averaged over the 3,354 violent events in our dataset—relative to movement that occurs in the

---

School of Information, University of California, Berkeley, CA, USA. ✉e-mail: jblumenstock@berkeley.edu

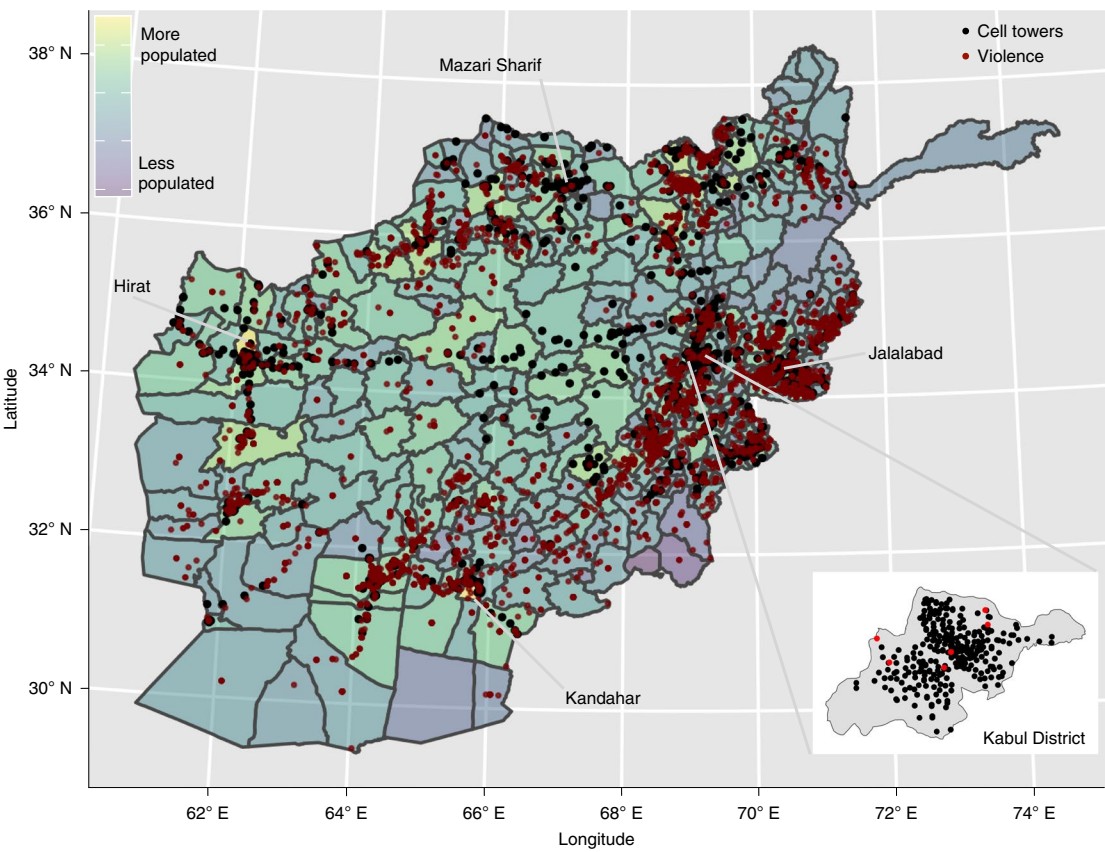

**Fig. 1 | Map of cell towers and violent events in Afghanistan, 2013–2017.** District boundaries are represented by grey lines. There are 1,439 cell tower groups (black dots) and 5,984 violent events (red dots)[75], 3,354 of which occur in districts on days with mobile phone activity. Event locations are marked by exact geocoordinates when available, or with the district centroid when only the district name is known.

absence of violence. This modelling approach and its identifying assumptions are described in detail in the Methods (section 'Panel regressions: measuring *k*-day displacement').

We frequently refer to this excess migration caused by violence as 'displacement', though we acknowledge that this statistical notion of displacement is different from that used by international organizations. For instance, the IOM defines displacement as "the movement of persons who have been forced or obliged to flee or to leave their homes or places of habitual residence, in particular as a result of or in order to avoid the effects of armed conflict, situations of generalized violence, violations of human rights or natural or human-made disasters."[30]

The results presented below illustrate the additional perspective on internal displacement that can be achieved through the analysis of population-scale digital trace data. However, this approach has important limitations. Some of these we can address (as discussed in the Methods), but others are more fundamental[21,31–34]. We discuss a few key limitations—such as issues of population representativity, data access and privacy—in the 'Limitations' section, after describing the empirical findings.

## Results

**The aggregate effect of violence on displacement.** Violence in Afghanistan causes internal displacement (Fig. 2). Among subscribers who were present on the day of a violent event, there is an immediate and statistically significant increase in the likelihood of leaving the district (Fig. 2a). This increase peaks roughly ten days after violence, when the odds of being observed in a different district are 4.0% higher than in the absence of violence (95% confidence

interval (CI), (2.6%, 5.5%); $P < 0.001$; this and all subsequent reported statistical tests are two-tailed tests). Violence-induced displacement is persistent: even 120 days after the violent event, subscribers who were present for the event are roughly 2% (95% CI, (0.5%, 2.9%); $P = 0.007$) more likely to still be outside of the district. We show that these results are robust to several potential data and modelling issues in the Methods, section 'Impact of a violent day'.

We also find evidence that displacement often anticipates violence. This is apparent in Fig. 2b, which shows the increase in the odds of an individual being in a different district from their location 30 days prior (Methods, section 'Panel regressions: measuring *k*-day displacement'). Roughly five days before violent events are reported in the media, subscribers start to leave the impacted regions. In the Discussion, we consider several possible explanations for this unexpected result.

**What types of violence cause the most displacement?** The aggregate effects shown in Fig. 2 mask substantial heterogeneity in how different types of violence cause different patterns of displacement. Most notably, violence involving the Islamic State (IS), while less frequent, causes significantly more displacement than violence involving the Taliban (Fig. 3a). The difference is the most pronounced in the immediate aftermath of the event, when the increase in displacement is ten percentage points higher for violence involving IS (for subscribers present during the violence, events involving IS increase the odds of displacement a day after by 12.7% (95% CI, (5.7%, 20.2%); $P < 0.001$); Taliban, 1.9% (95% CI, (−0.4%, 4.3%); $P = 0.111$); Taliban–IS difference in coefficient estimate, 0.10 (95% CI, (0.04, 0.17); $P = 0.002$)). Such evidence is consistent with the fact

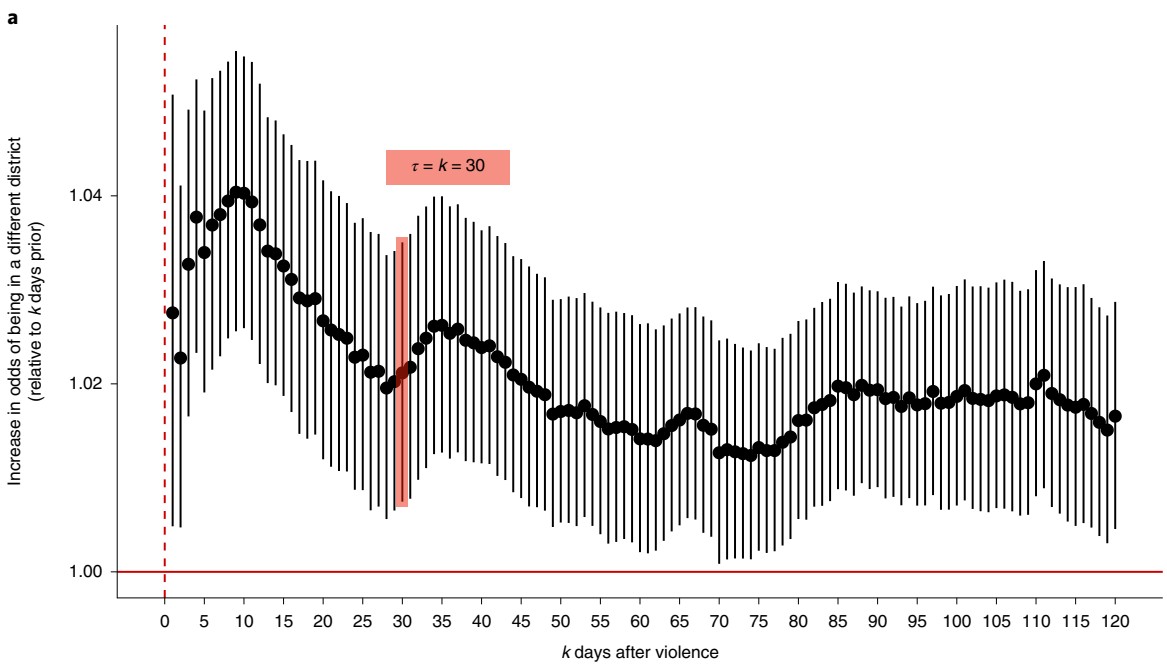

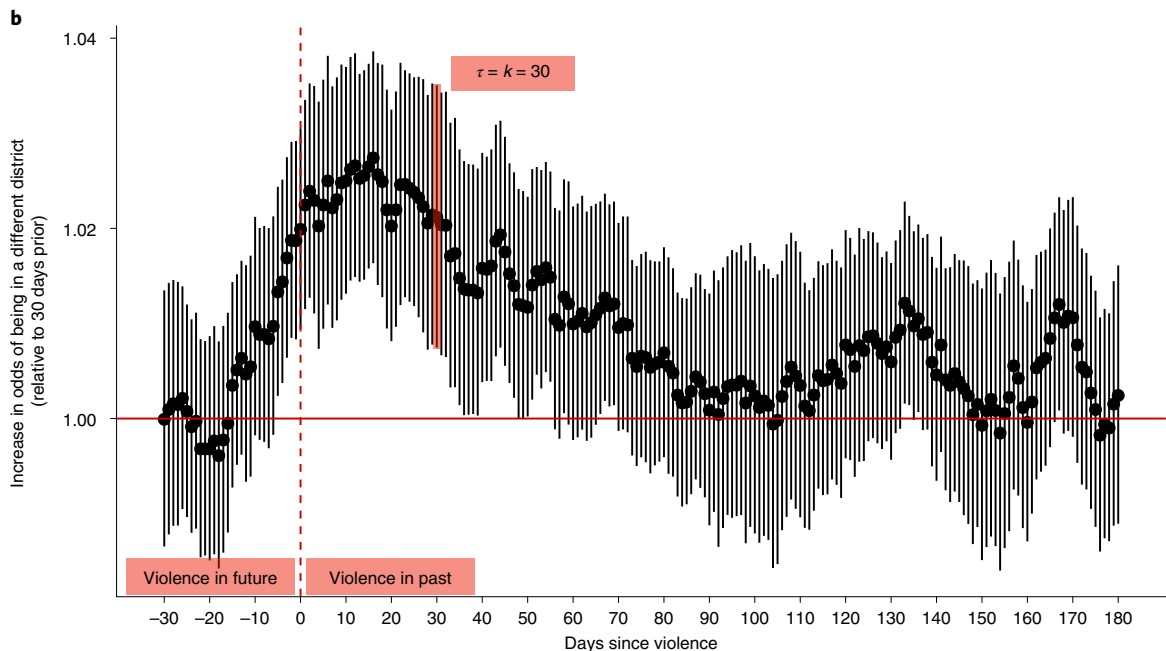

**Fig. 2 | The effect of violence on internal displacement. a**, Displacement for those present on the day of the violent event. Exponentiated regression coefficients, plotted on the y axis, indicate the increase in the odds that individuals who were in the district on the day of violence are in a different district $k$ days later. The bars indicate 95% CIs. The estimates are based on 3,354 violent events. $\tau$ indicates the number of lags for treatment variables in the regression model (Methods), representing the number of days since violence. The full results are provided in Supplementary Table 2. **b**, Thirty-day displacement. Exponentiated regression coefficients indicate the increase in the odds that individuals are in a different district than they were 30 days prior. Negative x-axis values correspond to days preceding violence. The bars indicate 95% CIs. The estimates are based on 3,354 violent events. $\tau$ represents days since violence. The full results are provided in Supplementary Table 3.

that IS attacks are particularly brutal and frequently target civilians; the Taliban have condemned IS attacks as 'heinous'[35,36].

High-casualty and high-frequency violence also have larger effects on displacement. Figure 3b shows that the highest-casualty events (11 or more casualties, roughly equivalent to the 10% of events with the most casualties) cause more displacement at all periods following the event, relative to lower-casualty events ($P < 0.001$

for a paired $t$-test of the 120 values shown in the figure; mean difference, 0.034; 95% CI, (0.031, 0.036)). Figure 3c indicates that violent events in regions that have recently experienced violence lead to more displacement than violence in regions that have experienced a period of relative peace ($P < 0.001$ for a paired $t$-test; mean difference, 0.030; 95% CI, (0.029, 0.031)). This result is perhaps surprising given prior work suggesting that people in chronically violent areas

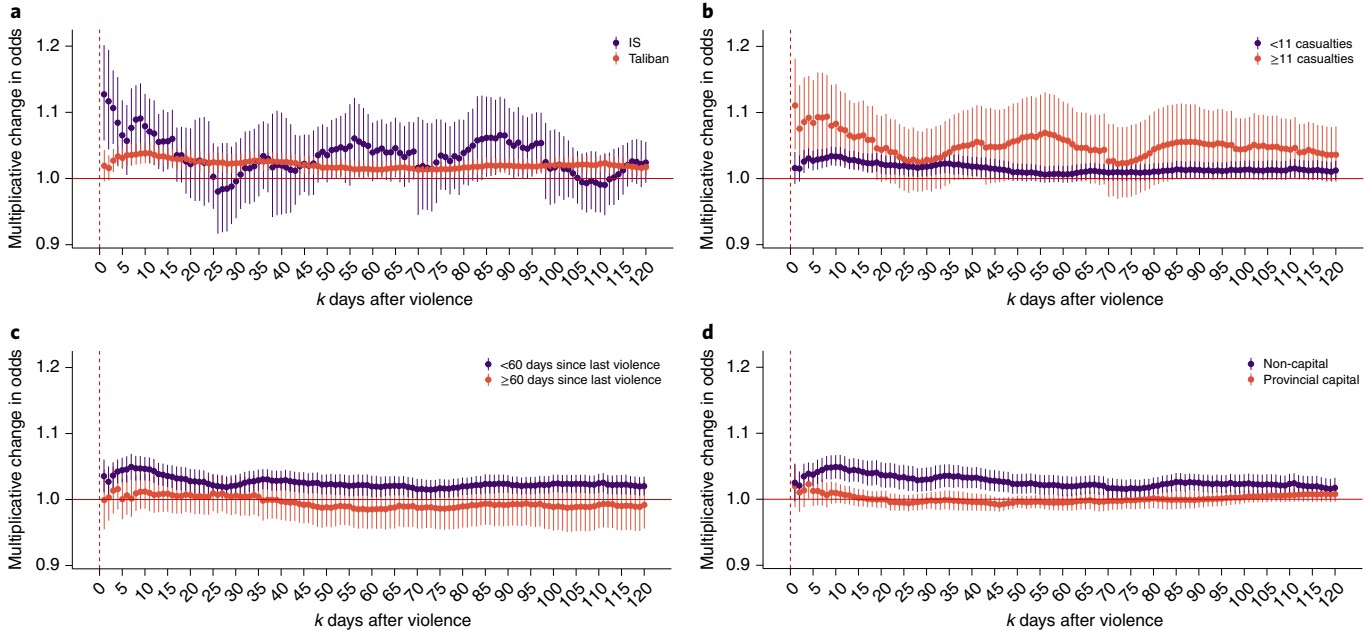

**Fig. 3 | The effect of violence on displacement, disaggregated by type and location. a–d,** The impact of a violent day on individuals who were in the district on the violent day. The coefficients are estimated separately for each panel. Panel **a** shows displacement for IS ($N=185$) versus Taliban violence ($N=3,134$); the full results are provided in Supplementary Table 4. Panel **b** shows displacement for events with 11 or more casualties ($N=397$) versus 10 or fewer casualties ($N=2,957$); the full results are provided in Supplementary Table 5. Panel **c** shows displacement for events following fewer than 60 days of peace ($N=2,319$) versus 60 or more days ($N=1,035$); the full results are provided in Supplementary Table 6. Panel **d** shows displacement for events in provincial capitals ($N=2,460$) versus non-capitals ($N=894$); the full results are provided in Supplementary Table 7.

may acclimate to conflict[37–39]. However, in the context of the prolonged and pervasive Afghan conflict, we take this as evidence that people are more likely to flee their homes after recent and sustained violence but may be willing to withstand 'idiosyncratic' violence.

We also find that the displacement response depends on whether the violence occurs in a provincial capital (which is typically an urban or peri-urban area) or a more rural non-capital district (Fig. 3d). Although provincial capitals appear relatively resilient to violence, with muted and largely insignificant increases in displacement, the effects of violence outside of provincial capitals are large and persistent ($P<0.001$ for a paired $t$-test; mean difference, 0.026; 95% CI, (0.024, 0.028)). Afghanistan's 34 provincial capitals are regional seats of government and are where the Afghan National Security Forces are typically concentrated[40]. As a result, they may be seen to offer relative safety compared with outlying districts. We also find suggestive evidence that displacement effects are the highest in areas that are under insurgent control or contested by the Taliban (Supplementary Fig. 3). We do not emphasize these results, since our data on insurgent control are not ideally suited for this analysis ('Limitations').

The heterogeneous displacement responses shown in Fig. 3a–d highlight how each separate characteristic of violent events—the parties involved, the severity and recency of violence, and the location of the attack—relate to subsequent displacement. However, some of these characteristics are correlated: for instance, violence in provincial capitals tends to be preceded by fewer days of peace. For this reason, Fig. 4 shows the joint relationship between these characteristics and displacement—that is, it shows how each factor correlates with displacement, holding the other factors fixed (using a regression model; Methods). Here we observe that the general patterns are consistent with the earlier results, but the outsized impact of IS is made clear: all else equal, violence involving IS has the largest and most pronounced impact on short- and long-term displacement. (For example, the difference in coefficient estimates for IS events versus events following recent violence, for the model involv-

ing the average effect of violence 1–15 days in the past, is 0.194 (95% CI, (0.032, 0.356); $P=0.019$).)

**Where do the displaced go?** The analysis of anonymized mobile phone metadata can also provide granular insight into the destinations of the displaced. To build intuition, Fig. 5 shows the flow of migrants in Afghanistan during normal times—that is, on days when violence does not occur. On such non-violent days, the total volume of mobile subscribers leaving capitals and non-capitals is approximately equal. For those moving from capitals, 73.5% move to a different province, and roughly half (47.0%) move to other capitals or major cities; of the subscribers leaving non-capitals, half of them (50.6%) move to another province, and 30.0% move to the provincial capital in the same province.

More revealing is how the equilibrium pattern of displacement shifts in response to violence. These results are summarized in Fig. 6, which shows how the odds of moving to each type of district change on days with violence. Violence affecting non-capital regions (left) makes subscribers more likely to go to the provincial capital of their origin district and less likely to go to the largest cities (Kabul, Kandahar, Hirat, Mazari Sharif and Jalalabad) outside their origin province. By contrast, violence affecting capital regions (right) tends to drive subscribers away from their home province and to either the five major cities or more rural non-capitals. We discuss these and related results below.

## Discussion

The preceding results illustrate a granular and dynamic empirical approach to studying conflict-induced displacement. The aggregate finding that violence causes displacement is consistent with prior work on internal displacement in fragile and conflict-affected countries[7,8,12]. In Afghanistan (the context of our analysis), IOM survey data from 2019 suggest that conflict had caused roughly two thirds of all current displacement[25].

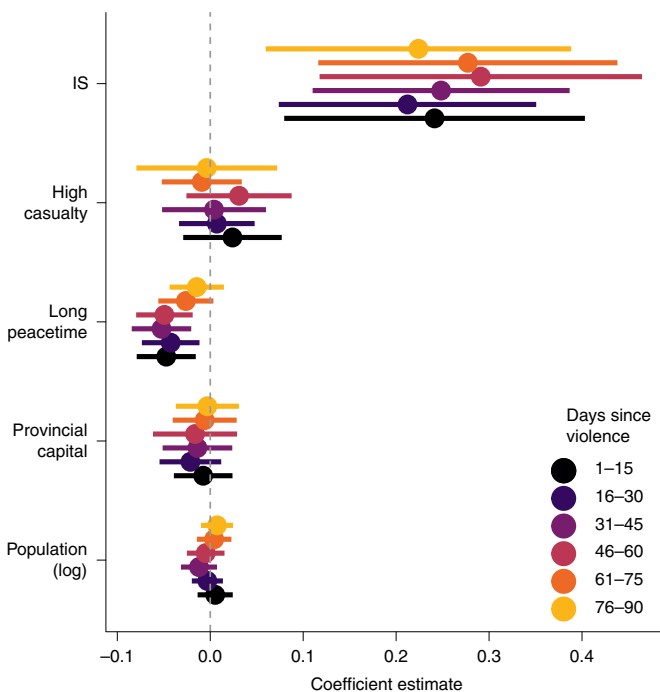

**Fig. 4 | How event and location characteristics affect the impact of each event on 30-day displacement (N = 2,359).** The different colours denote different outcome variables: the average effect of violence when it occurred 1–15, 16–30, 31–45, 46–60, 61–75 and 76–90 days in the past. The bars indicate 95% CIs. The full results are provided in Supplementary Table 8.

The main innovation of this approach is that it permits a fine-grained, quantitative analysis of violence-induced displacement that would be difficult to accomplish using household surveys and qualitative interviews. For instance, our analysis documents the important role of provincial capitals in influencing both the impact of violence and the destinations of displaced people. One specific finding is that when violence occurs in non-capitals, subscribers tend to flee to their home provincial capital. Such evidence corroborates qualitative findings that "people displaced by conflict and violence tended to try to stay as close as possible to their homes, moving from rural areas to the provincial capital or a neighbouring province"[41]. The attraction of provincial capitals is

probably due to several factors. First, capitals have a higher concentration of government security forces, and in the wake of violence, physical security is probably a crucial consideration. Relatedly, humanitarian aid—whose effectiveness depends on the security of the location[42]—is often most easily accessed in provincial capitals. Furthermore, provincial capitals are often the most urbanized area in the region, potentially offering greater economic opportunities[43]. Finally, movement to provincial capitals may create a feedback loop, where families are likely to have connections in provincial capitals, thus encouraging further movement to these capitals[26,44,45].

We also find that the types of violence associated with more displacement include violence related to IS, high-casualty violence and chronic violence. These results can be explained by the level of risk perceived by individuals and households influencing their decision to flee[7,8]. Apart from indiscriminate attacks, IS has orchestrated vivid displays of violence, such as filmed executions, advertising its brutality and intimidating opponents[46]. The Taliban, by contrast, has support among some segments of society and is seen by some as a legitimate governing force[47]. Separately, it is plausible that high-casualty and chronic violence similarly create an atmosphere of fear that leads to higher displacement. Casualties have been found to be associated with insurgent recruitment and violence[48]; their link to displacement is perhaps unsurprising.

Our analysis also indicates that people appear to anticipate the occurrence of violence, leaving before it occurs—a finding that relates to prior work on the predictability of violence and conflict[49–52], including recent work using mobile phone data[53]. This anticipatory effect is most pronounced with recently experienced violence (Supplementary Fig. 4). There are several possible related explanations for the anticipatory response we observe in Afghanistan. First, both NATO forces and the Taliban frequently warned civilians prior to major operations—for example, by distributing leaflets[54] or 'night letters'[55]. Relatedly, it may be that people were not anticipating a specific event but were rather responding to a general period of unrest; individuals might perceive a threat of violence before a recorded event actually takes place. (Note that this does not violate the causal identification assumption of no unobserved time-varying confounders, unless the perceived threat of violence causes violence to occur.) For example, there might be skirmishes between armed forces that do not lead to fatalities or are not reported in the media. This is consistent with survey-based evidence, which finds that the perceived threat of violence or presence of armed forces is sufficient to cause displacement, independent of the actual exposure to violence[7–9]. Rumours and word-of-mouth may also play a role: prior

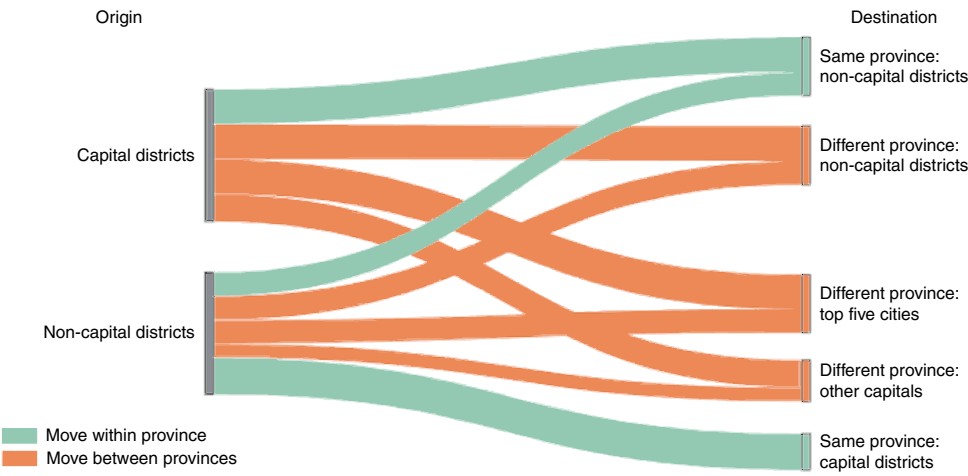

**Fig. 5 | Migrant flows on days without violence.** The proportion of subscribers moving between locations of different types, where a move is defined as a change of home district over a 30-day period.

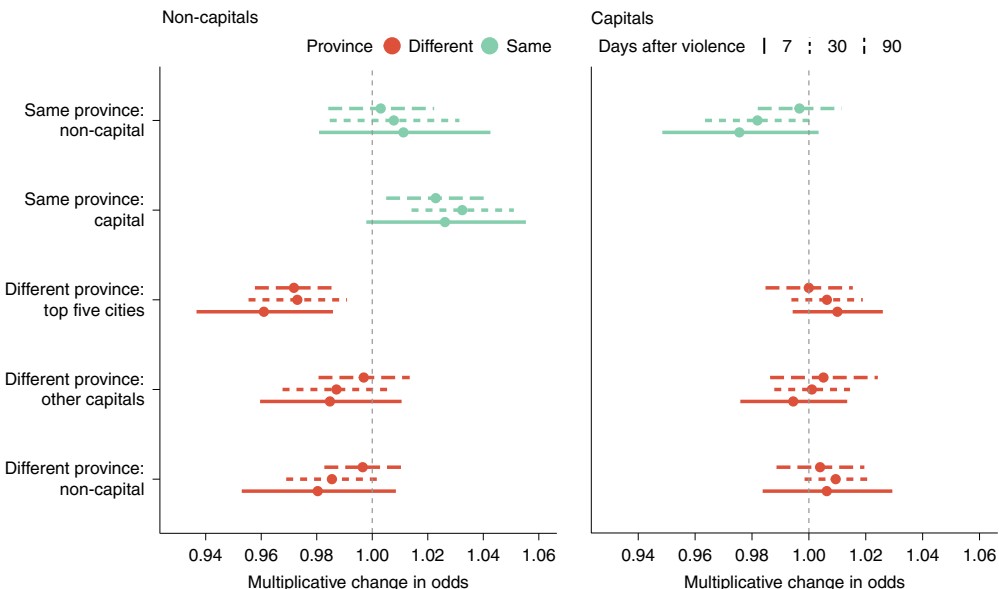

**Fig. 6 | The effect of violence on destination choice.** How movement from districts on days of violence differs from movement from districts on days without violence, where a move is defined as being in a different district 7, 30 or 90 days after the reference date (Methods). Left, movement from non-capitals; right, movement from capitals. Red denotes movement between provinces, and green denotes movement within a province. Destinations are divided by the type of district: one of the five largest cities (Kabul, Kandahar, Hirat, Mazari Sharif and Jalalabad), another of the provincial capital districts or a non-capital district. The bars indicate 95% CIs. The estimates are based on up to 894 violent days in non-capitals and 2,460 in provincial capitals. The full results are provided in Supplementary Table 9.

work suggests that information about violence spreads quickly[56,57]—including unverified rumours[58]—and that rumours may prompt people to take action to protect themselves[59].

**Limitations.** While passively collected mobile phone data create new possibilities for understanding forced displacement, there are important limitations and considerations regarding the use of such data (for more systematic reviews, see refs. [21,31–34]). As noted, our measures of migration and displacement only cover interdistrict movement of phones within the country of Afghanistan, and thus probably underestimate the overall effect of violence on displacement, which includes international and within-district movements.

More generally, there are several issues related to the representativity of the data and the potential biases that may result. Our data reflect the displacement patterns of mobile phone owners who have an active account on one specific commercial network, and not the full Afghan population. While mobility inferences on mobile phone owners have been shown to correlate with the mobility of non-owners in certain contexts[60], it is possible that violence induces different types of displacement among phone owners and non-owners. Specifically, the most vulnerable populations, such as women, children and lower-income individuals, tend to be underrepresented in mobile phone data, as they are less likely to own a phone[61,62]. Relatedly, the data indicate only the intermittent and approximate locations of mobile phones, not of actual individuals. When phones are shared, powered off or disused, this can introduce measurement error. Vulnerable populations may use their phones less often, resulting in larger errors for these individuals[63]. We address some of these issues through data processing, estimation methods and robustness checks (Methods and Supplementary Fig. 5) but cannot eliminate all such concerns.

Another obstacle to the widespread use of these methods of studying internal displacement is that mobile phone data are not always easily accessible and may require partnerships with data owners. However, there are encouraging signs that private companies (such as Facebook, Google and Safegraph[64,65]) may be interested

in making mobility data available to researchers and humanitarian organizations[31,66,67]. Finally, the analysis of phone data must respect the privacy of individual subscribers, particularly when dealing with displaced and otherwise vulnerable populations[68,69]. Our analysis involves only anonymized data that are aggregated geographically (by district) and temporally (by day). More generally, de Montjoye et al.[70], Mayer et al.[71] and others provide broader frameworks for the privacy-conscientious use of mobile phone data.

We also note limitations of the data we use to measure violence and conflict. We rely on data collected by the UCDP, which has known limitations that we discuss below. We also considered several alternative sources of data on violence, including the Armed Conflict Location and Event Data Project (ACLED)[72], the Global Terrorism Database (GTD)[73] and data on Significant Activities (SIGACTS)[74]. However, these alternative data sources either do not have full data available during our period of study (ACLED and SIGACTS) or focus on specific types of violence (that is, GTD focuses on terrorism). Thus, the UCDP data are the most suitable for our purposes, but they have their own limitations, in part because they are collected from media sources (media sources are supplemented by reports from non-governmental and intergovernmental organizations, field reports and books[75]). In particular, not all events are included in media reports; inclusion is affected by a large number of factors[76]. For example, more populous regions and economic centres have a larger media presence and are thus better covered than peripheral areas, while contested areas and places with less infrastructure tend to have poorer coverage. Media coverage is also influenced by how 'newsworthy' the event is likely to be[77]. Our analysis is further restricted to those events for which the date and location (that is, the district) are known. Since the spatial and temporal precision of event reporting is not random[78], and that non-randomness may even differ across regions of Afghanistan, this could introduce bias into our analysis. For instance, if the threshold for reporting violence in rural areas is higher than the threshold in urban areas[76,77], that could bias our analysis to finding larger effects of violence in rural areas. We perform robustness tests

in the Methods ('Implications of missing violence data') to assess the likely magnitude of this bias, but we acknowledge that this is a limitation of our data—and a limitation shared by most empirical work on conflict. While these problems of exclusion are faced by all violent events datasets that rely on news sources, and there is little we can do to improve the quality of the data, we are optimistic that the quality of conflict event data will improve as they gain more widespread use. For instance, several papers now articulate best practices for the collection of conflict data[79,80], and researchers are increasingly exploring how social media and other non-traditional data can be used to source conflict data[81].

Our analysis is further constrained by the general scarcity of publicly available quantitative data covering Afghanistan during this period. For instance, Supplementary Fig. 3 provides suggestive evidence that displacement effects are greatest when violence occurs in areas controlled or contested by the Taliban—a finding that is supported by prior work on the dynamics of civil wars[82]. However, we consider such results preliminary, because they are based on a cross-sectional assessment of territorial control conducted in October 2017. Since territorial control changed rapidly during the period we study, a more robust analysis requires panel data on territorial control over time.

A broader limitation of our quantitative approach is that we cannot say much about what specific aspects of violence people are reacting to and whether observed displacement is voluntary or not. In aggregating effects over all violent events, we find that, on average, violence causes displacement—but this may hide the fact that certain types of violence lead to a reduction in mobility (for instance, if populations are trapped or involuntarily immobile[83]), if that reduction is offset by other types of violence that dramatically increase out-migration. Relatedly, in discussing the anticipatory response, we noted that recorded (fatal) events may be preceded by a broader context of non-fatal violence, and individuals may in fact be responding to broader changes in the local environment that we cannot directly observe in our data.

Finally, while we have taken several steps to improve the internal validity of our analysis, we acknowledge that we can only speculate about the extent to which our findings will generalize to other countries. The general result that displacement is affected by violence has been documented in other contexts, including Colombia[8], Kosovo[10], Nepal[7], Somalia[11], Syria[9] and internationally[12]. However, the Afghan setting is unique in many respects, and some results (such as the magnitude of the estimated effect or specific elements of the heterogeneous response) may be specific to Afghanistan. For example, Afghanistan has a decades-long history of conflict; our results will probably translate better to settings with endemic conflict than ones where violence is new. Afghanistan also has strong clan-based, ethnic and sectarian divisions; a strong patriarchal society; and a relatively poor and young population. These contextual factors probably impact migration decisions in complex ways; our results will naturally be most relevant to environments with similar characteristics.

**Conclusion.** Despite these important caveats, we remain optimistic that mobile phone and other digital trace data offer great potential for the study of internal displacement. Our empirical analysis provides insight into the nature of violence-induced displacement in Afghanistan and helps quantify some of the human costs of violence that would be difficult to measure using traditional methods. While there are definite limitations to what can be observed through mobile phone data, conflict-prone regions are often also the places where traditional survey-based data are the least reliable and most difficult to obtain. Our hope is that this approach can complement traditional perspectives on displacement and eventually contribute to the design of effective policies for prevention and mitigation.

## Methods

**Data on violence in Afghanistan.** We obtain violent events data from the UCDP, a leading source of data on conflict events[72]. Specifically, we use UCDP Georeferenced Event Dataset Global version 19.1, available at https://ucdp.uu.se/downloads/. This open-source collection of metadata on armed conflict and organized violence is collected from media reports, so it is likely to be biased towards salient events in more populous regions[76,77]. The criteria for inclusion of an event are "the incidence of the use of armed force by an organized actor against another organized actor, or against civilians, resulting in at least one direct death in either the best, low or high estimate categories at a specific location and for a specific temporal duration"[75]. In Afghanistan from 2013 to 2017, 5,984 events were recorded where the event location is known to a district level and the event time is known to a specific day; 4,740 of these events occur during the period April 2013–March 2017, for which we have call detail records (CDR), and 3,354 of those occur in districts on days with mobile phone activity. We discard events that are not recorded with this level of precision (47% of all events; the potential implications of this are discussed in 'Limitations'). Afghanistan is divided into 398 districts in 34 provinces; our analysis is conducted on a district level.

**Mobile phone call detail records.** Our analysis of displacement is based on a large dataset of pseudonymized mobile phone data from one of Afghanistan's largest mobile phone operators. As described in 'Limitations', we take precautions to ensure that the analysis of phone data respects the privacy of individual subscribers. In particular, our analysis involves only pseudonymized data that are aggregated geographically (by district) and temporally (by day).

We obtain CDR that provide metadata for every mobile phone call and data packet transfer that occurred on this network from April 2013 to March 2017—a total of roughly 20 billion events. For each such event, we observe a pseudonymized unique identifier for the subscriber (hashed from their phone number), the date and time of the event, and the identifier of the physical mobile phone tower through which the transaction was routed. We also know the exact location of each tower, which allows us to approximately identify each subscriber's location at the time of the event, to within roughly 500 m in urban areas and roughly 10 km in rural areas.

There are 13,315 active towers during this period, many of which are very close together; we group these towers into 1,439 tower groups by combining towers less than 100 m apart. These cell tower groups are plotted in Fig. 1 and Supplementary Fig. 1. Only districts with cell towers are included in this analysis, though we note that these generally correspond to the more populated districts in Afghanistan (see Fig. 1).

**Measuring migration.** From the original CDR, we follow a sequence of steps to determine whether and when a migration event occurs. We adopt the IOM's definition of migration, which is "The movement of persons away from their place of usual residence, either across an international border or within a State"[30], and we focus on internal migration ("within a State"), where the place of usual residence is measured to a district-level precision. We capture trips that last approximately a week or more (at least five full days and two travel days). The migration that we measure is therefore an interdistrict movement. Complete details on this process are in the Supplementary Information (in the section on data processing); a brief summary is provided here.

Our first step is to derive a 'daily modal location' for each subscriber for each day, which is intended to capture the district in which the subscriber spends the majority of their time on that day. For each individual, we first compute their most commonly used cell tower in each hour. Then, for each 24-hour period from 06:00 to 06:00 the next day, we compute the mode of the hourly modal towers. The towers are then mapped to districts using point-in-polygon assignment. Similar methods have been used and validated in other work[13,15,32,84–87]. While several prior studies use night-time hours to infer daily locations, we instead use all hours, which allows us to include more individuals in our analysis. For example, in April 2013, data are available for approximately 31 million individual-days using night-time hours (18:00 to 06:00), while 61 million individual-days are defined when using all hours (06:00 to 06:00). The two approaches are highly correlated: of the 31 million observations available using night-time hours, 89% record the same daily modal districts when computed using all hours. Another common approach in the literature divides the physical terrain into approximate catchment areas of each cell tower, using a Voronoi tessellation (for example, refs. [13,88]). Our analysis focuses on slightly larger administrative districts, since many of our violent events are identified at only the district level.

In Afghanistan, we find that the geographic distribution of daily modal locations of mobile phone subscribers broadly reflects the geographic distribution of the population. In particular, when comparing the number of mobile phone users in each district to the district population as estimated by Afghanistan's Central Statistical Office, the Pearson's correlation coefficient is 0.94 (95% CI, (0.92, 0.95); $P < 0.001$). (We calculate the number of subscribers in each district as the number of subscribers whose daily modal location is assigned to that district, averaged over all days in which the district has a non-zero number of subscribers, for a one-year period. Official estimates are obtained from https://data.humdata.org/dataset/estimated-population-of-afghanistan-2015-2016. On a log scale, the Pearson's correlation is 0.53 (95% CI, (0.43, 0.61); $P < 0.001$).)

These daily modal locations tend to be sparse and noisy—for instance, many people do not use their phones on every single day, people may take short trips to nearby (non-residential) locations and so forth. Our second step thus employs an unsupervised scanning algorithm[29] to identify contiguous segments in which a subscriber is, with high probability, resident in a single district. This algorithm helps smooth the influence of noise (for instance, long periods when a person is primarily in one location but intermittently visits other locations for one or two days) and missing data (for instance, when a person uses their phone infrequently, but almost exclusively from the same location) and has the advantage of not arbitrarily grouping days into calendar weeks or calendar months.

The third step is to identify migration events using discontinuous breaks in these contiguous segments. The second and third steps use the open-source Python package migration_detector (https://github.com/g-chi/migration_detector), which is specifically designed to infer migration events in transaction log data. The accompanying paper[29] validates the use of these methods to measure migration. Tuning parameters are set to identify changes in locations resulting from stays of at least five full days in origin and destination districts. See the Supplementary Information for the full details.

The above procedures allow us to measure migration events from the mobile phone CDR. Many such events are not indicative of 'displacement', which the IOM defines as "The movement of persons who have been forced or obliged to flee or to leave their homes or places of habitual residence, in particular as a result of or in order to avoid the effects of armed conflict, situations of generalized violence, violations of human rights or natural or human-made disasters."[30] Given the limited contextual information available in the CDR, we cannot directly observe whether each inferred migration event should be considered a displacement. Instead, as we discuss in 'Panel regressions: measuring $k$-day displacement', we focus our analysis on the increase in out-migrations from a district that appear to be caused by violence in that district.

**Data validation.** To validate the measures of migration derived from the mobile phone CDR, we compare our derived migration metrics to displacement measures published by the IOM (DTM Afghanistan Districts Round 9 Baseline Assessment, available at https://data.humdata.org/dataset/afghanistan-displacement-data-baseline-assessment-iom-dtm). To our knowledge, there are no official or other published data measuring interdistrict migration as we do; while we try as far as possible to produce analogous measures, the IOM data measure fundamentally different quantities, and we do not expect comparisons to be identical. Generally speaking, we might expect province shares of migration and displacement to be similar if the fraction of displaced people among those who move for any reason is similar across provinces. This might not always be the case—for example, we might expect the capital, Kabul, to have a much smaller share of displaced people. Nevertheless, we make the comparison, as the IOM data are the closest published dataset on internal migration or displacement in Afghanistan.

The IOM collects data at the settlement (village) level through key informant interviews, focus group discussions and direct observation[25]. They use these data to estimate counts of outgoing and incoming internally displaced persons (IDPs) in assessed settlements over fixed periods. IDPs are categorized into 'returnee IDPs', 'arrival IDPs' and 'fled IDPs'. We use the data collected in the year 2016; to our knowledge, this means that these individuals were recorded as being IDPs anytime during the year. We group 'returnee' and 'arrival' IDPs together as incoming IDPs, treat 'fled IDPs' as outgoing IDPs and sum the total numbers of incoming and outgoing IDPs for each province. We then compute each province's share of the total incoming and outgoing IDPs.

Next, to construct an analogous metric from the CDR, we compare the district locations of each subscriber at the beginning and end of three four-month periods in 2016 (January–April, April–August and August–December, summed to obtain a measure of movement in 2016, since the longest we track subscribers is for 120 days). Since each district could have different cell-phone penetration rates, for each period and each district, we estimate the total number of people who moved in and out of the district by scaling the number of recorded subscribers who moved by $\frac{\text{district population}}{\text{no. of recorded subscribers}}$, where the district population is as estimated by Afghanistan's Central Statistical Office (available at https://data.humdata.org/dataset/estimated-population-of-afghanistan-2015-2016). We then aggregate these to the province level for 2016 and compute province shares in a similar manner. Supplementary Fig. 2a shows the share of each province estimated to leave; Spearman's correlation between CDR and IOM statistics at the province level is $\rho = 0.49$ (95% CI, (0.20, 0.72); $P = 0.004$). Supplementary Fig. 2b does the same for incoming individuals, with $\rho = 0.56$ (95% CI, (0.31, 0.77); $P < 0.001$).

In Supplementary Fig. 2, we see that many provinces have similar shares of migration and displacement, with some obvious differences in Kabul Province, where migration far exceeds displacement, and Hilmand, where displacement far exceeds migration (Hilmand Province was a Taliban stronghold and frequently saw heavy fighting[89]).

**Panel regressions: measuring $k$-day displacement.** We combine the violent events data and migrations observed in the CDR into a district-day panel dataset, which we use to estimate the 'average' impact of violence on out-migration from the district in which violence occurs. We estimate this effect by adapting widely used panel regression models to our context (for example, refs. [90–92]), which allows us to estimate the total migration caused by violence while controlling for unobserved district- and time-related factors that might influence the occurrence of both violence and migration. We first present the technical details of this model and later discuss the identifying assumptions and possible concerns with this approach.

For each value of $k$ from 1 to 120, we estimate the following regression:

$$g(\mathbb{E}(Y_{dt,k}|X_{dt}, T_{d,t+\tau})) = \gamma_d + \lambda_t + \sum_{\tau=-30}^{180} \beta_\tau T_{d,t+\tau} \qquad (1)$$

where $d$ indexes the district, $t$ indexes the time (calendar date) and covariates $X_{dt}$ are given by district fixed effects, $\gamma_d$, and time fixed effects, $\lambda_t$. $T_{d,t+\tau}$ are the 'treatment' variables (whether or not violence occurs) in district $d$ at time $t$, at a lag of $\tau$ days. Lags of $\tau \in [-30, 180]$ are used, representing violence in the district 30 days in the future to 180 days in the past. This range was chosen because all effects were observed to lie within this window; the results are insensitive to a longer window, while shorter windows are unable to capture all effects of interest. The outcome variable, $Y_{dt,k}$, is the proportion of those in district $d$ at time $t - k$ that are in a different district at time $t$. Subscribers present $k$ days ago in district $d$ but with a missing location on day $t$ are included in the denominator but not in the numerator in this computation. The parameter $k$ is introduced to capture the fact that displacement has to be measured relative to some time in the past. $g()$ is the logit link function. Since the outcome variable is a proportion, we model it using a beta distribution, a family of continuous distributions in the interval from 0 to 1, taking a variety of possible shapes depending on the values of its parameters. We fit a beta regression using maximum likelihood estimation[93]. Standard errors are clustered at the district level.

These coefficients can be interpreted as with a logistic regression: for each $\tau$, $e^{\beta_\tau}$ is the multiplicative change in the odds of being in a different district today (time $t$), for $T_\tau = 1$ (when violence occurs) relative to $T_\tau = 0$ (days without violence), holding the other variables constant. To interpret $\beta_\tau$ as the causal effect of violence on displacement, the target parameter is the causal conditional odds ratio, and the set of necessary identification assumptions are positivity, consistency, conditional exchangeability and correct model specification[94]. In our context, this specification assumes that there are no spatial spillovers, meaning that violence in one district does not have an effect on displacement in other districts. Carryover effects of the violence are limited to 180 days after the violence, and effects from up to 30 days prior are allowed. These daily effects are estimated independently and do not modify one another. The effects are assumed to be identical for all districts and to not vary over the measurement period (2013–2017). The confounders are limited to district, time and treatment in the surrounding window of time, and these enter additively. This implies that there are no unobserved time-varying confounders and that past outcomes do not affect current treatment (this is plausible since in most cases the number of displaced people is not large enough to affect military strategy). We relax several of these assumptions in subsequent analyses—for instance, by allowing for heterogeneous effects of different types of violence in different types of locations.

The key identifying assumption that there are no unobserved time-varying confounders requires the precise day in which a district experiences violence to be random, after conditioning on district and time fixed effects and the occurrence of violence in the surrounding window of time (equation (1)). Qualitatively, we find this assumption plausible because the precise timing (that is, the day on which violence occurs) of insurgent attacks is often meant to surprise government forces. However, the assumption cannot be tested directly; we therefore perform several checks to assess whether the occurrence of violence can be predicted beyond what our model in equation (1) captures. Specifically, we first regress the occurrence of violence on day $t$ on the control variables in equation (1)—that is, $g(\mathbb{E}(T_{dt}|\gamma_d, \lambda_t, T_{d,t+\tau})) = \gamma_d + \lambda_t + \sum_{\tau \in [-30,180] \setminus \{0\}} \beta_\tau T_{d,t+\tau}$—and obtain the residuals $T_{dt} - \hat{\mathbb{E}}(T_{dt})$. Supplementary Table 1 assesses whether these residuals can be predicted using recent lags and trends in the outcome variable (30-day displacement) and the number of subscribers observed to be in a district. We find that, using either a linear model or a machine learning approach (a random forest with tenfold cross-validation), these characteristics do not accurately predict residual violence ($R^2 \leq 0.00028$ (95% CI using non-parametric bootstrap, percentile interval, (0.00026, 0.0010))). Finally, as an additional robustness check, we find that adding more restrictive region × month time-varying fixed effects to equation (1) does not qualitatively change the main results (Supplementary Fig. 6).

In estimating these regressions, we exclude district-days in which the outcome variable is 0, 1 or missing. The rationale is that these zeros and ones are probably due to data sparsity. On one hand, if no subscribers were recorded as being in a different district, it could be that their locations were simply missing (for example, they did not use their phones, there was no cell service or they switched providers). On the other hand, it is unlikely that all subscribers would have left a district on any day; a recorded 1 could indicate cell tower outages in the origin district, for example. (News reports have described the Taliban restricting access to communications or destroying cell towers, and we do see a small reduction in the number of active cell towers in a district during periods of violence. However, we do not see significant decreases in call volumes at a district level, nor do we see a decrease in the probability of a district having an active tower, probably indicating

that individuals are able to connect to a different cell tower within the same district. If it is the case that all individuals are only able to connect to a cell tower in a different district, our response variable would be a 1 and hence dropped from the regression. This limits overestimation of the displacement response.)

Several other points are of note. First, this estimation of displacement as an increase in migration due to violence also partially addresses the concern that the place of usual residence might be incorrectly measured using CDR. If violence does not impact the measurement error (for example, if the likelihood of a subscriber being misallocated to the district of their workplace instead of their home does not change due to violence), then the misallocation will not bias the estimated displacement. Second, although the treatments (violent events) occur relatively infrequently, the statistical model we employ is robust to sparsity; if all of the events are recorded accurately, estimates will not be biased because of sparsity. Non-random missingness of recorded events could bias estimates and are discussed in 'Limitations'.

**Summary of identification strategy.** To more plainly summarize our statistical approach to measuring the effect of violence on displacement, we regress out-migration in each district-day on indicators for occurrences of violence up to 180 days prior and up to 30 days in the future, while controlling for geographic and temporal factors. This approach is designed to capture out-migration in excess of the out-migration that normally occurs in that district (on all other days) and on that day (in all other districts). Thus, the model does not assume that people do not move when violence is not occurring; instead, it uses movement in non-violent times and places as a baseline, to better isolate the *additional* movement that co-occurs with violence.

We include 180 lag terms and 30 lead terms to measure excess out-migration (again relative to normal out-migration) that occurs in the 180 days after violence and in the 30 days leading up to violence, as well as excess out-migration on the day of violence. Since violent events may be spatially and temporally correlated, a single observation (district-day) in the regression could have multiple violence indicators that are turned on; the migration dependent variable for that observation would thus contribute to the estimation of violence effects on all of the affected leads and lags.

Using this regression framework allows us to estimate the 'average displacement effect' of violence, averaged over the 3,354 violent events in our dataset that occur in districts on days with recorded mobile phone activity. For example, a coefficient of 0.03 on the indicator for violence at a lag of ten days can be interpreted as 'On average, violence occurring ten days prior increases the odds of migration out of a district by 3% (a multiplicative change of $e^{0.03} \approx 1.03$), holding all other variables constant.' This approach helps limit the extent to which any one specific event, which might have unusual characteristics or correlates, can influence our final results. For instance, if one violent event happened to occur on a day in which a certain district would have seen unusual out-migration even in the absence of violence, that single event would have a limited impact on our final estimates. The main concern is if violent events were systematically correlated with other unobserved factors—above and beyond the flexible spatial and temporal fixed effects that we control for in the regression.

**Impact of a violent day.** To distil the impact of a single violent day, for each $k \in [1, 120]$, we consider the coefficient for $T_\tau$ for $\tau = k$. This coefficient captures the effect of violence occurring at a $\tau$ day lag, on movement measured at time $t$, compared with district locations $k$ days ago. When $\tau = k$, the outcome variable is measured with respect to those in the district on the day of the violence. In this way, extracting the relevant coefficients from regressions where the outcome variable is different values of $k$ gives us the impact of a single violent day, on the subscribers in the district on that day. We demonstrate the robustness of these results to potential data issues, such as the presence of outliers, as well as modelling issues such as the inclusion of additional time-varying controls, in Supplementary Figs. 5 and 6.

**Heterogeneous effects.** To allow for the possibility that the displacement response may differ for different types of violence or for types of locations, the results of heterogeneous effects models are shown in Fig. 3. These results are estimated by creating separate treatment indicators for different types of events (for example, low-casualty versus high-casualty), which replace the treatment indicators in equation (1). For instance, letting $H_{d,t+\tau}$ denote the occurrence of high-casualty (>10 casualties) violence and $L_{d,t+\tau}$ denote the occurrence of low-casualty violence, we estimate:

$$g(\mathbb{E}(Y_{dt,k}|X_{dt}, H_{d,t+\tau}, L_{d,t+\tau}))$$
$$= \gamma_d + \lambda_t + \sum_{\tau=-30}^{180} \beta_{H,\tau} H_{d,t+\tau} + \sum_{\tau=-30}^{180} \beta_{L,\tau} L_{d,t+\tau} \qquad (2)$$

When analysing the heterogeneity of response by location (for example, for provincial capitals), we estimate prior regressions on the relevant subsets of the data—that is, by only including observations pertaining to provincial capitals.

**Controlling for multiple dimensions of heterogeneity.** To account for multiple dimensions of heterogeneity varying jointly, we analyse 30-day displacement by

first fitting equation (3) separately for each of the events, using ordinary least squares:

$$\log \left( \frac{Y_{dt,30}}{1 - Y_{dt,30}} \right) = \gamma_d + \lambda_t + \sum_{\tau=-30}^{180} \beta_\tau T_{d,t+\tau} + \epsilon_{dt} \qquad (3)$$

Here $T_{d,t+\tau}$ indicates a single event at a time (each treatment indicator indicates whether or not the specific event occurs at district $d$ at time $t$, at a lag of $\tau$ days). Only events in which all $\beta_\tau$ coefficients can be estimated are included, meaning that if the outcome variable is unavailable in any day that is 30 days preceding the event to 180 days after the event, it is not included in the analysis. This results in a total of 2,359 events being studied. For each included event, we take the mean of the estimated coefficients for $\beta_\tau$, for $\tau = 1$–15, 16–30, 31–45, 46–60, 61–75 and 76–90. We treat these as outcome variables and model each of these derived outcomes $O_i$ as

$$O_i = \beta_0 + \beta_1 \text{provCap}_i + \beta_2 \log(\text{population})_i$$
$$+ \beta_3 \text{IS}_i + \beta_4 \text{casualties11}_i + \beta_5 \text{peace60}_i + \epsilon_i \qquad (4)$$

where $i$ is the event, $\text{provCap}_i$ is a binary variable denoting whether the event occurs in a provincial capital, $\log(\text{population})_i$ is the log of the population of the district in which the event occurs (added as a control), $\text{IS}_i$ is a binary variable denoting whether the event involved IS, $\text{casualties11}_i$ is a binary variable denoting whether the event was associated with 11 or more casualties and $\text{peace60}_i$ is a binary variable denoting whether the event was preceded by 60 or more days of peace. Figure 4 shows the estimated coefficients for each of the outcomes.

**Destinations of displaced people.** To investigate where the individuals displaced by violence go, we first examine migrant flows during non-event days (Fig. 5) and event days (Supplementary Fig. 7). We consider all recorded moves in any 30-day period and split these into days on which violent events occurred at the start of the 30-day period ('event days') and those on which no events were recorded ('non-event days'). We repeat the following analysis for each. First, we categorize recorded moves as originating in either capital districts or non-capital districts. We then split destination districts into mutually exclusive categories by first recording whether they are in the same or a different province from the origin district; these destinations are then partitioned into three different types of districts—the major urban cities (Kabul, Kandahar, Hirat, Mazari Sharif and Jalalabad), other capital districts and non-capital districts.

To estimate the effect of violence on the destination of displacement, we use a similar setup as equation (1). Instead of the outcome variable being the fraction of the population that moved on day $k$, we use the fraction of movers (those in a different district at time $t$ compared with $k$ days ago) on day $k$ observed to be at specific types of destination districts, as described above. We use outcomes for $k = 7, 30, 90$ and fit separate regressions for provincial capitals, for non-capitals and for each outcome. As before, district-days in which the outcome variable is 0, 1 or missing are excluded from the analysis.

**Implications of missing violence data.** As discussed in 'Limitations', our analysis does not include violent events that are not associated with specific locations (that is, where we do not know the district in which the event occurred). This could introduce bias into our analysis if certain types of violence (with specific migration responses) are systematically more or less likely to have known locations. We therefore conduct additional analysis to determine whether the spatial precision with which an event is recorded is correlated with the magnitude of the displacement effect.

Specifically, using the same empirical approach described in 'Heterogeneous effects', we create separate treatment indicators for each of the three types of events that we use in our analysis, based on their available geographic precision: (1) events for which the exact location is known and coded, $A_{d,t+\tau}$ ($N = 1,698$); (2) events that occurred within a 25 km radius around a known point, $B_{d,t+\tau}$ ($N = 789$); and (3) events for which only the district is known, $C_{d,t+\tau}$ ($N = 969$). These replace the treatment indicators in equation (1):

$$g(\mathbb{E}(Y_{dt,k}|X_{dt}, A_{d,t+\tau}, B_{d,t+\tau}, C_{d,t+\tau}))$$
$$= \gamma_d + \lambda_t + \sum_{\tau=-30}^{180} \beta_{A,\tau} A_{d,t+\tau} + \sum_{\tau=-30}^{180} \beta_{B,\tau} B_{d,t+\tau} + \sum_{\tau=-30}^{180} \beta_{C,\tau} C_{d,t+\tau} \qquad (5)$$

The results, shown in Supplementary Fig. 8, indicate that the displacement response is very similar for violent events with these three different levels of spatial precision. There are small differences in the point estimates, but the general pattern of the response is unchanged, and the CIs of all three violence types overlap. Of course, this analysis does not eliminate the possibility that there might be a qualitatively different displacement response to violent events that are not recorded in our dataset (or for which district information is unknown). Unfortunately, we cannot directly test that concern, since we cannot estimate the displacement effect of violence when the location of the violence is not known.

**Reporting Summary.** Further information on research design is available in the Nature Research Reporting Summary linked to this article.

## Data availability

The mobile phone dataset contains detailed information on roughly 20 billion mobile phone transactions in Afghanistan. These data contain proprietary and confidential information belonging to a private telecommunications operator and cannot be publicly released. Upon reasonable request, we can provide information to accredited academic researchers about how to request the proprietary data from the telecommunications operator. With the telecommunications operator's permission, we can also provide district-level aggregate measures of migration for replication purposes to accredited academic researchers. All other data used in this paper are publicly available, and the sources have been listed in the text. They are UCDP Georeferenced Event Dataset Global version 19.1 (https://ucdp.uu.se/downloads/), Afghanistan district-level population data (https://data.humdata.org/dataset/estimated-population-of-afghanistan-2015-2016) and displacement measures published by the IOM (DTM Afghanistan Districts Round 9 Baseline Assessment, available at https://data.humdata.org/dataset/afghanistan-displacement-data-baseline-assessment-iom-dtm). Shape files for Afghanistan are available at https://esoc.princeton.edu/data/administrative-boundaries-398-districts.

## Code availability

All code to reproduce the findings of this study is available at https://github.com/Global-Policy-Lab/afghanistan-internal-displacement.

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

## Acknowledgements

We thank M. Callen, O. Dube, T. Ghani, S. Kakade and J. Shapiro for helpful conversations. G. Chi, R. Gajendragadkar, S. Mittal and M. Jalis provided excellent research assistance on this project. J.E.B. received support for this work from the National Science Foundation under grant nos IIS-1942702 and CCF-1637360 and from the Office of Naval Research under award no. N00014-17-1-2313. The funders had no role in study design, data collection and analysis, decision to publish or preparation of the manuscript. Any opinions, findings, and conclusions or recommendations expressed in this material are those of the authors and do not necessarily reflect the views of the National Science Foundation or the Office of Naval Research.

## Author contributions

X.H.T., S.M. and J.E.B. conceived and designed the experiments, performed the experiments and contributed materials or analysis tools. X.H.T. and S.M. analysed the data. X.H.T. and J.E.B. wrote the paper.

## Competing interests

The authors declare no competing interests.

## Additional information

**Correspondence and requests for materials** should be addressed to Joshua E. Blumenstock.

# Reporting Summary

Nature Research wishes to improve the reproducibility of the work that we publish. This form provides structure for consistency and transparency in reporting. For further information on Nature Research policies, see our Editorial Policies and the Editorial Policy Checklist.

## Statistics

For all statistical analyses, confirm that the following items are present in the figure legend, table legend, main text, or Methods section.

| n/a | Confirmed | |
|---|---|---|
| ☐ | ☒ | The exact sample size (*n*) for each experimental group/condition, given as a discrete number and unit of measurement |
| ☐ | ☒ | A statement on whether measurements were taken from distinct samples or whether the same sample was measured repeatedly |
| ☐ | ☒ | The statistical test(s) used AND whether they are one- or two-sided<br>*Only common tests should be described solely by name; describe more complex techniques in the Methods section.* |
| ☐ | ☒ | A description of all covariates tested |
| ☐ | ☒ | A description of any assumptions or corrections, such as tests of normality and adjustment for multiple comparisons |
| ☐ | ☒ | A full description of the statistical parameters including central tendency (e.g. means) or other basic estimates (e.g. regression coefficient) AND variation (e.g. standard deviation) or associated estimates of uncertainty (e.g. confidence intervals) |
| ☐ | ☒ | For null hypothesis testing, the test statistic (e.g. *F*, *t*, *r*) with confidence intervals, effect sizes, degrees of freedom and *P* value noted<br>*Give P values as exact values whenever suitable.* |
| ☒ | ☐ | For Bayesian analysis, information on the choice of priors and Markov chain Monte Carlo settings |
| ☒ | ☐ | For hierarchical and complex designs, identification of the appropriate level for tests and full reporting of outcomes |
| ☐ | ☒ | Estimates of effect sizes (e.g. Cohen's *d*, Pearson's *r*), indicating how they were calculated |

*Our web collection on statistics for biologists contains articles on many of the points above.*

## Software and code

Policy information about availability of computer code

| | |
|---|---|
| Data collection | All code to reproduce the findings of this study is available at https://github.com/shikharmehra/afghanistan-internal-displacement. |
| Data analysis | All code to reproduce the findings of this study is available at https://github.com/shikharmehra/afghanistan-internal-displacement. We use Python 3.9.1, R 4.1.2, and custom Python package migration_detector v0.1.2 (https://github.com/g-chi/migration_detector). No other non-standard libraries were used. |

For manuscripts utilizing custom algorithms or software that are central to the research but not yet described in published literature, software must be made available to editors and reviewers. We strongly encourage code deposition in a community repository (e.g. GitHub). See the Nature Research guidelines for submitting code & software for further information.

## Data

Policy information about availability of data

All manuscripts must include a data availability statement. This statement should provide the following information, where applicable:

- Accession codes, unique identifiers, or web links for publicly available datasets
- A list of figures that have associated raw data
- A description of any restrictions on data availability

The mobile phone dataset contains detailed information on roughly 20 billion mobile phone transactions in Afghanistan. These data contain proprietary and confidential information belonging to a private telecommunications operator, and cannot be publicly released. Upon reasonable request, we can provide information to accredited academic researchers about how to request the proprietary data from the telecommunications operator. With the telecommunication operator's permission, we can also provide district-level aggregate measures of migration for replication purposes to accredited academic researchers. All other

# Field-specific reporting

Please select the one below that is the best fit for your research. If you are not sure, read the appropriate sections before making your selection.

☐ Life sciences  ☒ Behavioural & social sciences  ☐ Ecological, evolutionary & environmental sciences

For a reference copy of the document with all sections, see nature.com/documents/nr-reporting-summary-flat.pdf

# Behavioural & social sciences study design

All studies must disclose on these points even when the disclosure is negative.

| | |
|---|---|
| Study description | This is a quantitative study using mobile phone metadata. |
| Research sample | The dataset is collected from subscribers of Afghanistan's largest mobile phone operator, who made at least one call or recorded data use in the period April 2013 to March 2017. This contains roughly 20 billion mobile phone transactions and were collected by a private telecommunications operator. This dataset is not representative of the full Afghan population (but is representative of subscribers on this network). |
| Sampling strategy | All available data are used. |
| Data collection | Mobile phone data were collected by Afghanistan's largest mobile phone operator using their proprietary systems. Violent events data were collected from media reports and publicly released by the Uppsala Conflict Data Program. |
| Timing | Data are available from April 2013 to March 2017. |
| Data exclusions | No data were excluded from the analysis. |
| Non-participation | No participants were involved the study. |
| Randomization | This is a study using observational data, and no randomization was done. |

# Reporting for specific materials, systems and methods

We require information from authors about some types of materials, experimental systems and methods used in many studies. Here, indicate whether each material, system or method listed is relevant to your study. If you are not sure if a list item applies to your research, read the appropriate section before selecting a response.

## Materials & experimental systems

| n/a | Involved in the study |
|---|---|
| ☒ | ☐ Antibodies |
| ☒ | ☐ Eukaryotic cell lines |
| ☒ | ☐ Palaeontology and archaeology |
| ☒ | ☐ Animals and other organisms |
| ☒ | ☐ Human research participants |
| ☒ | ☐ Clinical data |
| ☒ | ☐ Dual use research of concern |

## Methods

| n/a | Involved in the study |
|---|---|
| ☒ | ☐ ChIP-seq |
| ☒ | ☐ Flow cytometry |
| ☒ | ☐ MRI-based neuroimaging |

