## [Peer Review File. · Nature Human Behaviour]

Peer Review Information

Journal: Nature Human Behaviour

Manuscript Title: Joshua E. Blumenstock

Corresponding author name(s): Mobile Phone Data Reveal the Effects of Violence on Internal Displacement in Afghanistan

Editorial Notes:

Redactions Parts of this Peer Review File have been redacted as indicated to maintain third party confidentiality.

Reviewer Comments & Decisions:

Decision Letter, initial version:

21st July 2021

Dear Professor Blumenstock,

Thank you once again for your manuscript, entitled "Violence and Internal Displacement: Insights from Nationwide Mobile Phone Data", and for your patience during the peer review process.

Your Article has now been evaluated by 3 referees. You will see from their comments copied below that, although they find your work of considerable potential interest, they have raised quite substantial concerns. In light of these comments, we cannot accept the manuscript for publication, but would be interested in considering a revised version if you are willing and able to fully address reviewer and editorial concerns.

We hope you will find the referees' comments useful as you decide how to proceed. If you wish to submit a substantially revised manuscript, please bear in mind that we will be reluctant to approach the referees again in the absence of major revisions. We are committed to providing a fair and constructive peer-review process. Do not hesitate to contact us if there are specific requests from the reviewers that you believe are technically impossible or unlikely to yield a meaningful outcome.

To guide the scope of the revisions, the editors discuss the referee reports in detail within the team, including with the chief editor, with a view to (1) identifying key priorities that should be addressed in revision and (2) overruling referee requests that are deemed beyond the scope of the current study. We hope that you will find the prioritised set of referee points to be useful when revising your study. Please do not hesitate to get in touch if you would like to discuss these issues further.

In particular, your revision must address the following (as well as all other reviewer comments):

- 1) Address Reviewer 1's concerns about threats to inference related to the lack of a baseline mobility measure and the difficulty of isolating the effects of particular events from overall levels of violence.
- 2) Address the concerns raised by all 3 reviewers about the volume of data exclusions and possible missing data.
- 3) Ensure that your study follows standard practices for mobile phone data analyses, as highlighted by Reviewer 3. Please include validation of the user base and ensure that any deviations from standard practices are thoroughly explained and justified.
- 4) Include a transparent discussion of limitations. The reviewers have highlighted a number of important limitations of your study, and these must be thoroughly and transparently discussed in your manuscript.

Finally, your revised manuscript must comply fully with our editorial policies and formatting requirements. Failure to do so will result in your manuscript being returned to you, which will delay its consideration. To assist you in this process, I have attached a checklist that lists all of our requirements. Please note that, although Reviewer 1 suggested that re-ordering the sections would improve the clarity of your manuscript, we are bound by strict formatting requirements. We therefore must ask that you keep the Methods section at the end of your paper. If you have any questions about any of our policies or formatting, please don't hesitate to contact me.

If you wish to submit a suitably revised manuscript we would hope to receive it within 4 months. We

understand that the COVID-19 pandemic is causing significant disruptions which may prevent you from carrying out the additional work required for resubmission of your manuscript within this timeframe. If you are unable to submit your revised manuscript within 6 months, please let us know. We will be happy to extend the submission date to enable you to complete your work on the revision.

- Include a "Response to the editors and reviewers" document detailing, point-by-point, how you addressed each editor and referee comment. If no action was taken to address a point, you must provide a compelling argument. This response will be used by the editors to evaluate your revision and sent back to the reviewers along with the revised manuscript.
- Highlight all changes made to your manuscript or provide us with a version that tracks changes.

[REDACTED]

We look forward to seeing the revised manuscript and thank you for the opportunity to review your work. Please do not hesitate to contact me if you have any questions or would like to discuss these revisions further.

Sincerely,
Aisha

Aisha Bradshaw, PhD
Senior Editor
Nature Human Behaviour

Reviewer expertise:

Reviewer #1: political science, civil conflict

Reviewer #2: political science, civil conflict

Reviewer #3: network science, mobile phone data analysis

REVIEWER COMMENTS:

Reviewer #1

Key results: Please summarise what you consider to be the outstanding features of the work.

This manuscript examines the effect of violence on internal displacement in Afghanistan from 2013-2017. It confirms that violence causes internal displacement, and it adds nuance to how this

relationship works. In particular, civilians both anticipate and react to violence, IS attacks cause the most internal displacement, civilians tend to move to provincial capitals, and violence in provincial capitals is less likely to trigger internal displacement than violence outside provincial capitals.

I should note that the biggest strengths of this work are in methodological novelty and the evidence-based advance. My comments suggest ways to improve engagement with existing research and add nuance and clarity to the findings.

Validity: Does the manuscript have flaws which should prohibit its publication? If so, please provide details.

I am very supportive of this manuscript. I do not see prohibitive flaws. Instead, there are some points for revision and clarification.

Originality and significance: What are the major claims of the paper? Do you think that they represent a significant advance in the field? If the conclusions are not original, please provide relevant references. On a more subjective note, do you feel that the results presented are of immediate interest to many people in your own discipline, and/or to people from several disciplines?

The paper does not conflict with existing findings in the migration field, but I don't think that reduces the paper's value. Instead, it is fascinating to see that this unique data resource yields findings that are consistent with existing research that uses very different data and methods. The nuance about the large effect of IS violence is interesting, as is the authors' ability to highlight the role of provincial capitals. It is also valuable that the authors can show both how civilians anticipate and react to violence.

On a subjective note, I do think that the results are of great interest across disciplines. Political science pays insufficient attention to migration and displacement, so the paper may have greater impact in other disciplines, but that is a failing of political science as a discipline, not of the quality or importance of the authors' work. Economics, sociology, anthropology, and the interdisciplinary fields of international studies and migration studies should all be interested in this work. Regional experts are also likely to be interested in the nuances of the Afghan case that the authors discuss.

Data & methodology: Please comment on the validity of the approach, quality of the data and quality of presentation. Please note that we expect our reviewers to review all data, including any extended data and supplementary information. Is the reporting of data and methodology sufficiently detailed and transparent to enable reproducing the results?

This is where the greatest strengths of the paper exist.

Preregistration: If any part of the work reported in the manuscript was pre-registered, did the authors follow their preregistration plan? Did they report any deviations from their preregistration? Note that we ask authors to provide a link to the pre-registration in the Methods section and state the date of pre-registration. We also ask that authors disclose all deviations from the pre-registered protocol and explain the rationale for deviation (e.g., flaw, suboptimality, or reviewer/editorial request). In cases of deviation from the analysis plan, the originally planned analyses need to be reported in Supplementary Information.

Not applicable.

Appropriate use of statistics and treatment of uncertainties: Please include in your report a specific comment on the appropriateness of any statistical tests, and the accuracy of the description of any error bars and probability values.

Statistical tests appear appropriate and are transparent about uncertainties.

Custom code: If the work includes custom code, does the code run as intended? If you are unable to access the code, please contact us.

This part does not appear fully relevant to this manuscript, since it uses proprietary data. There is code that is publicly available for Python, though my Python skills are insufficient to properly test it out right now. I will be interested to try out their code as my Python skills improve.

Conclusions: Do you find that the conclusions and data interpretation are robust, valid and reliable?

I trust the author's conclusions, and I think the authors have done very careful empirical work.

Suggested improvements: Please list additional analyses, experiments or data that could help strengthening the work in a revision.

Given that mobile phone data is a measure of communication by definition, I'm very curious about whether the authors can show any information transmission mechanisms for their argument. Do call volumes change before or after violence? If a given violent event is motivating migration, I would expect call volumes to rise immediately after that violent event as people

discuss with each other what happened and what to do about it. That could also allow the authors to specify how exactly civilians are anticipating violence.

Since violence is spatially and temporally autocorrelated, how can the authors specify that migration is responding to a specific violent event? Where they observe migration as an anticipation of violence, how can they tell that civilians are not instead reacting to previous violent events? It appears to be a serious limitation that the authors cannot distinguish responses to specific events vs. responses to a broader context of violence or dangerous shifts in conflict dynamics. I'm not sure if the authors can resolve this, so if they can't then they need to explicitly mention this limitation in the text. This challenge is an issue that Schon (2015) and Schon (2019), which the authors cite, acknowledge and grapple with.

I would like to see at least a brief mention of whether there are any concerns about cell towers getting destroyed or damaged during violent periods. Could it be a threat to measurement if armed groups damage cell towers, thereby blocking cell phone signals and causing a notable drop in calls that can be completed? The issue of conflict destroying infrastructure is discussed in a wide variety of research, such as Schultz & Mankin (2019) on how weather stations get damaged or destroyed in African conflict zones.

By using mobile phone data, the authors are measuring the mobility of people. It is unclear which movements are displacement and which movements are part of normal mobility patterns. This is the biggest challenge with the data. Not all movement is displacement, and the authors do not have pre-conflict data to use as a source of comparison to help them distinguish displacement from mobility. The authors need to do more to engage with this challenge. It looks like they are assuming that mobility patterns that suddenly change are displacement, but this assumption likely over-classifies movement as displacement. Is there some way for the authors to identify what a "normal mobility pattern" looks like? Then, the authors could describe how a different mobility pattern can be interpreted as displacement.

The approach that the authors use is an interesting attempt to get at this idea. Yet, their approach assumes away the possibility that people will move when violence is not occurring. This is a potentially serious problem, since there is research showing that violence can deter movement (see work on involuntary immobility or trapped populations). Civilians may wait until the real or proverbial smoke clears before they move. I'm willing to be convinced that this challenge cannot be fully solved, but the authors would then need to discuss in the main text that future research could build on this by explicitly measuring internal displacement immediately following violence *and* internal displacement that occurs after some delay.

The analysis would benefit from considering the role of territorial control. There is available data on whether the government or the Taliban controls a piece of territory from the Special Investigator General for Afghanistan Reconstruction in their quarterly reports. It is not a perfect resource, but it can give the authors an indication of whether violent events are occurring in government-controlled territory. The consequences of violent events are likely to be different depending on which armed group controls that territory.

References: Does this manuscript reference previous literature appropriately? If not, what references should be included or excluded?

The authors should consider involuntary immobility, and how that may threaten their findings. In this respect, I suggest the following articles as useful to consider:

Lubkemann, S. (2008). "Involuntary Immobility: On a theoretical invisibility in forced migration studies." Journal of Refugee Studies **21**(4): 454-475.

Schon, J. (2016). "The centrality of checkpoints for civilians during conflict." Civil Wars **18**(3): 281-310.

When discussing the ability of civilians to anticipate violence, the authors may benefit from the following:

Rubin, J. and W. Moore (2007). "Risk Factors for Forced Migrant Flight." Conflict Management and Peace Science **24**(2): 85-104.

Uzonyi, G. (2014). "Unpacking the Effects of Genocide and Politicide on Forced Migration." Conflict Management and Peace Science **31**(3): 225-243.

Specifically, when discussing civilian threat perceptions, the authors may benefit from acknowledging the role of rumors in conflict zones:

Allport, G. and L. Postman (1947). The Psychology of Rumor. New York, Russell & Russell.

Greenhill, K. M. and B. Oppenheim (2017). "Rumor has it: The adoption of unverified information in conflict zones." International Studies Quarterly: sqx015.

Larson, J. M. and J. I. Lewis (2018). "Rumors, Kinship Networks, and Rebel Group Formation." International Organization: 1-33.

Schon, J. (2020). "How narratives and evidence influence rumor belief in conflict zones: Evidence from Syria." Perspectives on Politics: 1-14.

Silverman, D., Kaltenthaler, K., & Dagher, M. (2021). Seeing Is Disbelieving: The Depths and Limits of Factual Misinformation in War. International Studies Quarterly.

Clarity and context: Is the abstract clear, accessible? Are abstract, introduction and conclusions appropriate?

The abstract is clear. There are just unnecessary commas after the word events and before the word cause. My only comment here is that the authors may want a plainer way to call provincial capitals "resilient to local violence." Resilience has a ton of meanings across academic disciplines and topics, so I think it would be better to use different language there.

In the opening paragraph of the Introduction, the estimate that two-thirds of the global extreme poor will be living in conflict and fragile situations is a bit odd. I don't think it helps the authors. The first half of this sentence is fine, but the prediction by 2030 implies a lot of things about the future of conflict that I don't think the authors want to get into. There is no cost to them of just dropping this.

I'm not sure if the journal has requirements influencing this, but I found the sections out of order for how I read manuscripts. I would prefer to see the Materials and Methods discussion come before Results and Discussion. This is a simple switching of the order of sections, so it is an easy edit.

Please indicate any particular part of the manuscript, data, or analyses that you feel is outside the scope of your expertise, or that you were unable to assess fully.

I think it would be valuable for experts in big data and machine learning methods, ideally with experience working with mobile phone data, to review this paper. I have read the paper with an eye towards its contributions to understanding conflict-induced migration. From that perspective, I am excited about this paper. I learned a lot from it.

Reviewer #2:

Remarks to the Author:

Overall I think this is an excellent manuscript, it makes an important contribution and is competently executed. Before recommending publication however, I would like to see the authors address three issues (in falling order of importance).

First, I would like for the authors to address the loss of 47% of the violent events in the UCDP data. The authors do an admirable job of discussing and convincingly addressing a range of other inferential problems, but the potential threats to valid inference posed by this non-random loss of data are not addressed. It should be, because it is at least possible that it affects the findings. The reason they discard 47% of the violent events is because they are insufficiently precise with regards either to location or day. Here is how this might affect the findings and how we understand them:

Precision is going to be related to the core – periphery dimension, including presence/absence of cell towers and capital vs noncapital districts (Cf Weidmann 2014).

The threshold for reporting (esp with sufficient detail to be coded with enough precision to be included in the present study) is lower in core areas, and higher in peripheral areas (Cf Galtung & Ruge 1965, Öberg & Sollenberg 2011).

So what are some of the things that might lift an event in a peripheral area over the threshold and increase the likelihood that it is reported (with sufficient precision)? Events that represents a new development in a conflict, a break with a previous pattern, such as when control of an area changes hands, or when a new (and esp news worthy) actor makes advances/establish themselves or become active in new areas (Cf Öberg & Sollenberg 2011).

Thus, of the reported events with sufficient precision to be included you are likely to have a higher proportion of "business as usual"/typical events in capital districts (and other more core areas) and a higher proportion of events associated with new and more threatening developments in peripheral areas. The bar for something being reported (esp with geographic precision) is higher in peripheral areas, so there will in peripheral areas be a selection towards more newsworthy events like the appearance of new

actors, the territorial advance/retreat of some actor, or the loss of control by the government (or the seeming threat thereof) – all of which may imply higher levels of threat than otherwise equivalent violence and higher levels of threat may generate more displacement.

This might explain or be part of the explanation for the findings in Figure 3d provincial capitals vs non-capitals. It could possibly effect other findings also, like those on IS and on previous violent events, but that is less obvious.

While there is little that can be done about the lack of geographic precision, I think the authors need to take serious potential problems it raises and at least discuss potential implications.

To understand the problem I suggests the authors consult the sources below (they also give references to potential further readings).

On how reporting is related to core-periphery using military SIGACTS data for Afghanistan and comparing it to an earlier version of the UCDP data:

Nils B. Weidmann 2014. "On the accuracy of Media Based Conflict Event Data" *Journal of Conflict Resolution* 59(6): 1129-1149 <https://doi.org/10.1177/0022002714530431>

On how news evaluation works and might affect what gets reported and from where (i.e. what is newsworthy in different contexts):

Galtung, Johan, Ruge, Mari Holmboe. 1965. "The Structure of Foreign News: The Presentation of the Congo, Cuba and Cyprus Crises in Four Norwegian Newspapers." *Journal of Peace Research* 2 (1): 64–91.
Öberg, Magnus, Sollenberg, Margareta. 2011. "Gathering Conflict Information Using News Resources." In *Understanding Peace Research: Methods and Challenges*, edited by Höglund, Kristine, Öberg, Magnus, 47–73. New York: Routledge.

Second, scope conditions. How far do the findings in this manuscript travel? Are the findings particular to, or contingent on some aspect of this context, or can we expect similar patterns to obtain elsewhere? As a conflict country Afghanistan may in some ways be an outlier. Afghanistan have a 40+ year history of war and displacement. Millions of people have left, and it seems likely that 40 years of fighting has led to some social adaptations and strategies for coping with violence and threats of violence that are not present in other contexts. They have several ethnic and sectarian divisions layered on top of a partly disrupted clan based society all of which is reflected in the armed groups. And there is much foreign involvement and interference. So, for example, does the tendency to flee to district capitals have something to do with clan and ethnic or sectarian divisions (where one can safely flee), or is it because of safety in numbers, or is it because of stronger government control/presence in the district capitals as the authors suggest? Presumably the idp's seek some measure of safety, but government control does not always imply safety, even if it means less fighting. Does the age structure affect the propensity to flee? The median age in Afghanistan is just over 18 years and it is one of the poorest countries on the planet. Would patterns be similar in a country with a different age structure and more resources? These are not the only questions and I do not know if they are necessarily the most salient ones so I do not expect answers to them specifically. But I think some mention or discussion of what the authors think about the scope conditions of their findings are in order. It is valuable for future research. It is also, I think, important for ethical reasons to at least flag clearly if the authors are not able to delineate the scope conditions so that policy makers and others who may act of these findings in other settings are aware that they may not apply - at least not equally - in other cases.

Finally, the subject of this study is the displacement of Afghan people in Afghanistan in the internationalized but still very much Afghan civil war. The war in Afghanistan pre-dates the current US intervention by a quarter of a century or so. Yet, in the abstract a justification for looking at Afghanistan

seems to be that it is the longest war in US history. While this may be true, it strikes me as irrelevant to the study and bizarrely US-centric. I would suggest the authors simply drop that part.

Reviewer #3:

Remarks to the Author:

In this paper, the authors use mobile phone traces to measure internal displacement within Afghanistan in response to violent events. The authors are able to quantify where people are displaced to and for how long they remain displaced with respect to their home location. All locations are defined to district (admin 2) level. The authors report that there is a causal and significant increase in out-displacement following (and to a lesser extent preceding) a violent event. The nature of the displacement is affected by the group responsible for the attack, the severity and frequency of the violence as well as the proximity to urban/rural areas.

I believe that this paper is a notable and high quality contribution to the scientific literature on conflict, migration and the use of non-traditional data sources for official statistics. I would also add that this work is not a pure academic work but closer to 'scholar-practice' and so it should be reviewed with that in mind.

What this means concretely is the following; a large part of the contribution here is not concerned with methodological advancement but with the enormous work that goes into a complex partnership with a mobile phone operator in a low income country. To navigate the regulations of accessing this data in a responsible way, dealing with data cleaning and bringing to publication the study of such a sensitive topic is quite an achievement. Part of the value is the precedent that this sets for further similar studies and opportunities for other researchers and partnerships and hopefully, eventually, the operationalisation of this kind of work into sustainable partnerships between governmental organisations and mobile phone operators.

The flipside of this is that this study will surely be read by many technically literate and skeptical non-academics who work in National Statistical Agencies and similar. As a result, speaking from deep experience, any far reaching claims made by the authors could potentially be damaging to this endeavour in the long term.

With this in mind, my main substantive request to the authors is to be more clear about the limitations of this work. This is not a criticism of the work itself, it is simply the nature of the events and the data.

-- Figure 1 is the most critical of the entire paper: the spatial distribution of cell towers and violent events. Anyone who works with mobile phone data knows that cell towers are heterogeneously distributed and it will surprise no-one that there are parts of Afghanistan with no cell phone coverage. So please break out this figure into 3 parallel maps showing (i) cell tower locations (ii) violent events and (iii) district population so that these can be easily compared.

-- It is common practice to evaluate the user base in mobile phone studies by a simple comparison of district population from official statistics and mobile subscriber home location. Of course the best official statistics in Afghanistan will be imperfect but this kind of validation is important and will increase the impact of the work in the long term.

-- Again, common practice in mobile phone studies is to construct a Voronoi tessellation of cell towers and to use the overlap of the Voronoi cells with the administrative districts to, perhaps probabilistically, assign a home location. It's not clear to me from the provided details and citation (A general approach to

detecting migration events in digital trace data) what the authors did instead and why. Some more explanation here would be welcomed.

-- An overarching issue in studies of conflict and other extreme events is that they are very sparse. The case of Afghanistan is no different, by my estimate there are roughly $365 \times 4 \times 398 \sim 600\text{k}$ district-days in the data, with $\sim 3\text{k}$ violent events with district information giving a frequency of $\sim 0.5\%$. This may have some consequences for the validity of the causal analysis, I'm not qualified to comment on that, however at the very least the authors should make some reference to this aspect of the data and what confounding effects it might introduce.

-- Relatedly, the authors report that only 47% of the ground truth data on violent events have district level spatial accuracy. Again, this is no fault of the authors but should be addressed clearly. In fact, the use of passively collected data for proxying measures of interest is increasingly held back by a lack of high quality data against which to validate. The authors should comment on (i) what the elimination of roughly half the violent events might mean for their results and (ii) how these shortcomings in conflict data might be addressed in the future. I can recommend the following paper as an entry point into this debate; Growing pains for global monitoring of societal events (2016), Wang et al (<https://science.sciencemag.org/content/353/6307/1502.summary>)

Finally, I would be interested in the authors perspective on the degree to which this is a descriptive analysis vs a predictive endeavour. I would imagine that there is a delay in reporting and compiling violent events in the Uppsala database and similar. Would it be possible to use mobile phone derived measures of displacement as an early warning of a violent event (particularly in light of the previous points regarding sparsity and false positives)?

Author Rebuttal to Initial comments

R1's comments are in black/Arial; our point-by-point responses are in blue/Times

Thank you for your extremely thoughtful and constructive review, and for the generally enthusiastic assessment of our manuscript. We have provided a point-by-point response to each of your Suggested Improvements below.

You may also notice other changes to the manuscript, some of which were done in response to comments from the editor and other reviewers, and others were implemented to meet strict formatting requirements of the journal (including changes to the title and abstract, and the removal of footnotes and italics). We think the end result is a much-improved manuscript, though at times we struggled to balance the desire to thoroughly address reviewers' comments with the space constraints of the journal.

Thanks again for taking the time to review this manuscript!

Comment 1:

Given that mobile phone data is a measure of communication by definition, I'm very curious about whether the authors can show any information transmission mechanisms for their argument. Do call volumes change before or after violence? If a given violent event is motivating migration, I would expect call volumes to rise immediately after that violent event as people discuss with each other what happened and what to do about it. That could also allow the authors to specify how exactly civilians are anticipating violence.

This is an interesting point. We have conducted basic analysis that suggests that there is, on average, a spike in call volumes on the day of violence. This is consistent with your suggestion that call volume rises immediately after violence as people discuss what happened and what to do. The figure below shows the results of a regression analogous to what we use in the manuscript (Equation 1). The response variable is the number of calls recorded per district day, and we use leads and lags of up to 180 days.¹ The interpretation of each plotted coefficient is “the multiplicative increase in number of calls made in the district, on average, when violence occurs in the district at lag t , holding all other variables constant.” The coefficient significantly larger than 1 on the day of the violence indicates that call volume is more than 2.5% higher, on average, on days in which violence occurs, compared to non-violent days.

This line of analysis is something that has been explored in a few papers that we are aware of, such as Bagrow et al. 2011 and Gao et al. 2014. After careful consideration, we

¹ We use a log transformation on the response variable (the number of calls is extremely right-skewed), fit the model using ordinary least squares, with standard errors clustered at a district level. We include district-days with one or more calls.

elected to not incorporate this new analysis into the manuscript, as we were concerned that it might distract from our main focus, which is the effect on out-migration and displacement. However, we now briefly note that this response exists, and point the reader to prior work that explores the question in greater depth. The relevant text reads, “Rumors and word-of-mouth may also play a role: prior work suggests that information about violence spreads quickly (Bagrow et al. 2011, Gao et al. 2014) -- including unverified rumors (Silverman et al. 2021) -- and that rumors may prompt people to take action to protect themselves (Schon 2021). Consistent with this idea, we do observe a significant increase in call volume on days on which violence occurs.”

(We revisit the issue of rumors in response to your comment #9 below.)

Comment 2:

Since violence is spatially and temporally autocorrelated, how can the authors specify that migration is responding to a specific violent event? Where they observe migration as an anticipation of violence, how can they tell that civilians are not instead reacting to previous violent events? It appears to be a serious limitation that the authors cannot distinguish responses to specific events vs. responses to a broader context of violence or dangerous shifts in conflict dynamics. I’m not sure if the authors can resolve this, so if they can’t then they need to explicitly mention this limitation in the text. This challenge is an issue that Schon (2015) and Schon (2019), which the authors cite, acknowledge and grapple with.

Thank you for raising this important point. In the revised manuscript, we are now much more explicit about what concerns our modeling strategy allows us to address, and what limitations still remain.

Specifically, we use a panel event study model, which takes into account spatially and temporally autocorrelated violence by including leads and lags for violence indicators (up to 180 days in the past and 30 days into the future). Thus, the analysis is essentially asking, “On average, are more people leaving a particular district on a particular day that violence occurs *relative to the number who leave that district on a non-violent day, controlling for other violence that occurs in the 180 days before and 30 days after?*” In this way, we are estimating the idiosyncratic increase in out-migration associated with each specific violent event, averaged across all violent events. It may be the case that one single event was preceded by several violent events, and so for that event our estimate of “day of” displacement could capture some of the displacement from the prior events. But after incorporating displacement data covering all violent events in the country, our analysis tells us what the “average” displacement effect of violence is, on the day of violent events, as well as in the 180 days prior and 30 days post.

We have reorganized the paper to try to clarify these points. Specifically, the *Introduction* now provides more details on our modeling strategy: “Finally, we use a high-frequency panel event study design to estimate the causal effect of violence on migration, which we measure as the increase in movement of subscribers out of a district impacted by violence, relative to movement from the same district on non-violent days, while controlling for seasonality and other temporal factors. Specifically, we regress total out-migration from each district on each day on a vector of binary indicators for whether violence occurred in that district on that day, in the 180 days prior (the ‘lags’) or the 30 days after (the ‘leads’) that day. We control for district and day fixed effects, and cluster standard errors at the district level. This approach allows us to estimate the “average displacement effect” of violence -- averaged over the 5,984 violent events in our dataset -- relative to displacement that occurs in the absence of violence. This modeling approach and its identifying assumptions are described in detail in *Materials and Methods*, Section E.”

In addition, we have added a new subsection, “Summary of Identification Strategy”, to *Materials and Methods* Section E, which is our attempt to succinctly describe what our model is able to estimate. The text of this section is reproduced below:

To more plainly summarize our statistical approach to measuring the effect of violence on displacement: We regress out-migration on each district-day on indicators for occurrences of violence up to 180 days prior, and up to 30 days in the future, while controlling for geographic and temporal factors. This approach is designed to capture out-migration in excess of the out-migration that normally occurs in that district (on all other days) and on that day (in all other districts). Thus, the model does not assume that people do not move when violence is not occurring; instead, it uses movement in non-violent times and places as a baseline, to better isolate the *additional* movement that co-occurs with violence.

We include 180 lag terms and 30 lead terms in order to measure excess out-migration (again relative to normal out-migration) that occurs in the 180 days after violence, and that occurs in the 30 days leading up to violence, as well as excess out-migration on the day of violence. Since violent events may be spatially and temporally correlated, a single observation (district-day) in the regression could have multiple violence indicators that are turned on; the migration dependent variable for that observation would thus contribute to the estimation of violence effects on all of the affected leads and lags.

Using this regression framework allows us to estimate the “average displacement effect” of violence, averaged over the 5,984 violent events in our dataset. For

example, a coefficient of .03 on the indicator for violence at a lag of 10 days can be interpreted as "on average, violence occurring 10 days prior increases the odds of migration out of a district by 3% (a multiplicative change of $e^{.03} \sim 1.03$) holding all other variables constant." This approach helps limit the extent to which any one specific event, which might have unusual characteristics or correlates, can influence our final results. For instance, if one violent event happened to occur on a day in which a certain district would have seen unusual out-migration even in the absence of violence, that single event would have limited impact on our final estimates. The main concern is if violent events were systematically correlated with other unobserved factors -- above and beyond the flexible spatial and temporal fixed effects that we control for in the regression.

More broadly, however, it is possible that unobserved factors in the environment (such as tension, or even violence that occurs but does not result in a fatality) could precede or co-occur with the violent events that we do observe. In this case, we would be mis-attributing the displacement to the observed violent events, rather than the unobserved contextual factors. In addition to the above text (and the "main concern" noted in the last sentence quoted above), we now acknowledge this issue more explicitly in the *Limitations* section as follows:

A broader limitation of our quantitative approach is that we cannot say much about what specific aspects of violence people are reacting to, and whether observed displacement is voluntary or not. In aggregating effects over all violent events, we find that, on average, violence causes displacement -- but this may hide the fact that certain types of violence lead to a reduction in mobility (for instance, if populations are trapped or involuntarily immobile (Lubkemann 2008)), if that reduction is offset by other types of violence that dramatically increase out-migration. Related, in discussing the anticipatory response, we noted that recorded (fatal) events may be preceded by a broader context of non-fatal violence, and individuals may in fact be responding to broader changes in the local environment that we cannot directly observe in our data.

Comment 3:

I would like to see at least a brief mention of whether there are any concerns about cell towers getting destroyed or damaged during violent periods. Could it be a threat to measurement if armed groups damage cell towers, thereby blocking cell phone signals and causing a notable drop in calls that can be completed? The issue of conflict destroying infrastructure is discussed

in a wide variety of research, such as Schultz & Mankin (2019) on how weather stations get damaged or destroyed in African conflict zones.

Thank you for raising this interesting point. We have conducted additional analysis to explore this in more detail. This analysis suggests that (i) there is, on average, a small decrease in the number of active cell towers in the period surrounding violence; but (ii) in general, single tower outages are not severe enough to reduce coverage at the district level.

Specifically, we include two figures below, both of which are generated using regressions analogous to what we use to report our main results in the manuscript (Equation 1). In the first figure, the response variable is the number of active towers in a district on a given day.² The second figure uses a binary response variable that indicates whether a district has at least one active tower each day.³ Both figures include leads and lags of up to 180 days.

² We fit the model using ordinary least squares, with standard errors clustered at a district level. District-days in which there are 1 or more active cell towers are included.

³ Here we include all district-days for districts with at least one active cell tower during our measurement period (2013-2017). We fit a linear probability model using ordinary least squares, with standard errors clustered at a district level.

The first figure indicates that, starting roughly two months before fatal violence, there is, on average, a slight decrease in the number of active towers. This suggests that cell towers might be destroyed or turned off prior to recorded fatal events. News reports have suggested that a tactic the Taliban employs to restrict communications is to ask mobile phone operators to turn off coverage, or destroy towers, which is consistent with these results.⁴ There are several possible implications of this reduction in active cell towers: (1) individuals connect to a different cell tower within the same district, (2) individuals connect to a cell tower in a different district, and (3) individuals are unable to get service. How would these impact our analysis? If it is just (1), that should not affect our results, since all of our analysis is done at the district level. (2) is more of a concern, since it could upward bias our estimates of out-migration. However, Afghan districts are quite large, and towers are frequently redundant within districts (particularly in more populous areas). And as can be seen in the second figure, violence does not significantly impact whether the district as a whole has one or more active towers. (Also note that if *all* individuals are only able to connect to towers in a different district, those observations get discarded, and would not influence our results.) Finally, for (3), when individuals are able to reconnect to a cell tower within the district, they will be recorded after they are able to reconnect, resulting in a period with no data. In our data processing (described in *Materials and Methods* Section C), sparse segments in which individuals have no sustained activity in a different district are interpolated, so they will be recorded in the same district, and this is unlikely to bias results.

In our revised manuscript, we now provide a brief summary of this analysis, which reads, “News reports have described the Taliban restricting access to communications or

⁴ <https://www.voanews.com/extremism-watch/taliban-warns-phone-companies-shut-down-their-coverage-ghazni>, <https://www.nytimes.com/2019/10/02/world/asia/afghanistan-taliban-cell-towers.html>

destroying cell towers, and we do see a small reduction in the number of active cell towers in a district during periods of violence. However, we do not see significant decreases in call volumes at a district level, nor do we see a decrease in the probability of a district having an active tower, likely indicating that individuals are able to connect to a different cell tower within the same district. If it is the case that all individuals are only able to connect to a cell tower in a different district, our response variable would be a 1 and hence dropped from the regression. This limits overestimation of the displacement response.”

Comment 4:

By using mobile phone data, the authors are measuring the mobility of people. It is unclear which movements are displacement and which movements are part of normal mobility patterns. This is the biggest challenge with the data. Not all movement is displacement, and the authors do not have pre-conflict data to use as a source of comparison to help them distinguish displacement from mobility. The authors need to do more to engage with this challenge. It looks like they are assuming that mobility patterns that suddenly change are displacement, but this assumption likely over-classifies movement as displacement. Is there some way for the authors to identify what a “normal mobility pattern” looks like? Then, the authors could describe how a different mobility pattern can be interpreted as displacement.

We apologize for the lack of clarity in how we previously described our modeling approach. Your comments have helped us understand how confusing that description was, and (we hope) have enabled us to write a more complete and intuitive version. In the text quoted above (in our response to Comment #2; from *Materials and Methods* Section E), we have tried to address this concern. As noted there, it is not just sudden changes in mobility that we count as displacement, but rather changes in mobility in excess of the “normal mobility pattern” that you describe. The model defines “normal” as the mobility typical of each district (across the entire period for which we have mobility data), after accounting for what is typical on each day (across all districts in Afghanistan).

Comment 5:

The approach that the authors use is an interesting attempt to get at this idea. Yet, their approach assumes away the possibility that people will move when violence is not occurring. This is a potentially serious problem, since there is research showing that violence can deter movement (see work on involuntary immobility or trapped populations). Civilians may wait until the real or proverbial smoke clears before they move. I’m willing to be convinced that this challenge cannot be fully solved, but the authors would then need to discuss in the main text

that future research could build on this by explicitly measuring internal displacement immediately following violence and internal displacement that occurs after some delay.

We believe this comment follows directly from Comments #2 and #4, and again results from our lack of clarity in describing our modeling strategy. We hope the additional explanations, discussed above, help address this concern. But just to restate: we do not assume that people do not move when violence is not occurring. Rather, the statistical model, by way of the district fixed effect, uses the movement out of each district on all days in the dataset (the vast majority of which are non-violent) as a baseline to determine the excess (unusual) movement that occurs on days with violence.

You also raise the valid point that violence can deter movement, and cause involuntary immobility. It is certainly possible that this occurs in Afghanistan, though unfortunately the data we have do not make it easy to investigate this hypothesis. In essence, our analysis says that, on average, out-migration increases in times of violence. However, it is possible that some violence deters migration, but those events are more than offset by other violence that increases migration. In the revision, we more clearly acknowledge this limitation in the new *Limitations* section, the relevant text of which is reproduced here: “A broader limitation of our quantitative approach is that we cannot say much about what specific aspects of violence people are reacting to, and whether observed displacement is voluntary or not. In aggregating effects over all violent events, we find that, on average, violence causes displacement -- but this may hide the fact that certain types of violence lead to a reduction in mobility (for instance, if populations are trapped or involuntarily immobile (Lubkemann 2008)), if that reduction is offset by other types of violence that dramatically increase out-migration.”

Comment 6:

The analysis would benefit from considering the role of territorial control. There is available data on whether the government or the Taliban controls a piece of territory from the Special Investigator General for Afghanistan Reconstruction in their quarterly reports. It is not a perfect resource, but it can give the authors an indication of whether violent events are occurring in government-controlled territory. The consequences of violent events are likely to be different depending on which armed group controls that territory.

Thank you for this excellent suggestion. We have looked into the SIGAR reports, and our understanding is that SIGAR began assessing territorial control at a district level in December 2015. Unfortunately, the quarterly reports that we have been able to obtain

only include a summary and not a classification of individual districts, for example: ‘USFOR-A reports that approximately 70.5% of the country’s districts were under Afghan government control or influence as of January 29, 2016. Of the 407 districts within the 34 provinces, 287 districts were under government control or influence, 26 districts (6.4%) within 11 provinces were under insurgent control or influence, and 94 districts (23.1%) were “at risk.”’ The only available map and list of districts we have found is from October 2017, and was provided as an addendum to the 2018 Q1 report (https://www.sigar.mil/pdf/quarterlyreports/Addendum_2018-01-30qr.pdf). These data are not ideal, as territorial control changes over time, but we nonetheless use them to assess whether districts that were under Government vs. Taliban control in 2017 have different responses to violence. Empirically, we use a regression specification similar to the one used to compare provincial capitals vs. non-capitals (see *Materials and Methods* Section G).

The figure below shows the results for N=1469 events that occurred in districts under government control or influence vs. N=1885 events in districts that are contested or under insurgent control or influence (i.e., districts where SIGAR indicates insurgent control, insurgent influence, or contested control). We see a larger displacement response in districts under Taliban control/contested.

In the manuscript, we briefly describe these results in the main text as follows, “We also find suggestive evidence that displacement effects are highest in areas that are under insurgent control, or contested by the Taliban (*SI Appendix Figure S2*). We do not emphasize these results, since our data on insurgent control are not ideally suited for this analysis (see *Limitations*).” Note that we do not feature these results more prominently because we have concerns about the underlying data, which we discuss in the *Limitations* section as follows:

Our analysis is further constrained by the general scarcity of publicly available quantitative data covering Afghanistan during this period. For instance, *SI Appendix Figure S2* provides suggestive evidence that displacement effects are greatest when violence occurs in areas controlled or contested by the Taliban -- a finding that is supported by prior work on the dynamics of civil wars (Esparza et al. 2020). However, we consider such results preliminary, because they are based on a cross-sectional assessment of territorial control conducted in October 2017. Since territorial control changed rapidly during the period we study, a more reliable analysis would require panel data on territorial control over time.

Comment 7:

The authors should consider involuntary immobility, and how that may threaten their findings. In this respect, I suggest the following articles as useful to consider:

Lubkemann, S. (2008). "Involuntary Immobility: On a theoretical invisibility in forced migration studies." *Journal of Refugee Studies* 21(4): 454-475.

Schon, J. (2016). "The centrality of checkpoints for civilians during conflict." *Civil Wars* 18(3): 281-310.

Thank you for pointing this out. We have included this in the *Limitations* section -- we are unable to distinguish involuntary immobility from voluntary immobility, any involuntary immobility will result in an underestimation of the effect of violence on displacement.

Comment 8:

When discussing the ability of civilians to anticipate violence, the authors may benefit from the following:

Rubin, J. and W. Moore (2007). "Risk Factors for Forced Migrant Flight." *Conflict Management and Peace Science* 24(2): 85-104.

Uzonyi, G. (2014). "Unpacking the Effects of Genocide and Politicide on Forced Migration." *Conflict Management and Peace Science* 31(3): 225-243.

Thank you for these suggestions. We have expanded the discussion of anticipatory effects, and included one of the suggested references.

Comment 9:

Specifically, when discussing civilian threat perceptions, the authors may benefit from acknowledging the role of rumors in conflict zones:

Allport, G. and L. Postman (1947). *The Psychology of Rumor*. New York, Russell & Russell.
Greenhill, K. M. and B. Oppenheim (2017). "Rumor has it: The adoption of unverified information in conflict zones." *International Studies Quarterly*: sqx015.

Larson, J. M. and J. I. Lewis (2018). "Rumors, Kinship Networks, and Rebel Group Formation." *International Organization*: 1-33.

Schon, J. (2020). "How narratives and evidence influence rumor belief in conflict zones: Evidence from Syria." *Perspectives on Politics*: 1-14.

Silverman, D., Kaltenthaler, K., & Dagher, M. (2021). *Seeing Is Disbelieving: The Depths and Limits of Factual Misinformation in War*. *International Studies Quarterly*.

Thank you for these suggestions. The revised manuscript now contains a discussion of the potential role of rumors, which reads, "Rumors and word-of-mouth may also play a role: prior work suggests that information about violence spreads quickly (Bagrow et al. 2011, Gao et al. 2014) -- including unverified rumors (Silverman et al. 2021) -- and that rumors may prompt people to take action to protect themselves (Schon 2021). Consistent with this idea, we do observe a significant increase in call volume on days on which violence occurs."

Comment 10:

The abstract is clear. There are just unnecessary commas after the word events and before the word cause. My only comment here is that the authors may want a plainer way to call provincial capitals "resilient to local violence." Resilience has a ton of meanings across academic disciplines and topics, so I think it would be better to use different language there.

Thank you for this suggestion. We have removed this part of the sentence (due to word limits on the abstract), which now reads, "Provincial capitals act as magnets for people fleeing violence in outlying areas."

Comment 11:

In the opening paragraph of the Introduction, the estimate that two-thirds of the global extreme poor will be living in conflict and fragile situations is a bit odd. I don't think it helps the authors.

The first half of this sentence is fine, but the prediction by 2030 implies a lot of things about the future of conflict that I don't think the authors want to get into. There is no cost to them of just dropping this.

Thank you for pointing this out. We have removed the second half of that sentence.

Comment 12:

I'm not sure if the journal has requirements influencing this, but I found the sections out of order for how I read manuscripts. I would prefer to see the Materials and Methods discussion come before Results and Discussion. This is a simple switching of the order of sections, so it is an easy edit.

Thank you for this suggestion -- we are unable to make this change due to the journal's formatting requirements.

R2's comments are in black/Arial; our point-by-point responses are in blue/Times

Thank you very much for your positive assessment of our manuscript, and for providing very clear and constructive feedback. You will find point-by-point response to each of your Suggested Improvements below.

You may also notice other changes to the manuscript, some of which were done in response to comments from the editor and other reviewers, and others were implemented to meet strict formatting requirements of the journal (including changes to the title and abstract, and the removal of footnotes and italics). We think the end result is a much-improved manuscript, though at times we struggled to balance the desire to thoroughly address reviewers' comments with the space constraints of the journal.

Thanks again for taking the time to review this manuscript!

Comment 1:

Overall I think this is an excellent manuscript, it makes an important contribution and is competently executed. Before recommending publication however, I would like to see the authors address three issues (in falling order of importance).

First, I would like for the authors to address the loss of 47% of the violent events in the UCDP data. The authors do an admirable job of discussing and convincingly addressing a range of other inferential problems, but the potential threats to valid inference posed by this non-random loss of data are not addressed. It should be, because it is at least possible that it affects the findings. The reason they discard 47% of the violent events is because they are insufficiently precise with regards either to location or day. Here is how this might affect the findings and how we understand them:

Precision is going to be related to the core – periphery dimension, including presence/absence of cell towers and capital vs noncapital districts (Cf Weidmann 2014).

The threshold for reporting (esp with sufficient detail to be coded with enough precision to be included in the present study) is lower in core areas, and higher in peripheral areas (Cf Galtung & Ruge 1965, Öberg & Sollenberg 2011).

So what are some of the things that might lift an event in a peripheral area over the threshold and increase the likelihood that it is reported (with sufficient precision)? Events that represents a new development in a conflict, a break with a previous pattern, such as when control of an area changes hands, or when a new (and esp news worthy) actor makes advances/establish themselves or become active in new areas (Cf Öberg & Sollenberg 2011).

Thus, of the reported events with sufficient precision to be included you are likely to have a higher proportion of “business as usual”/typical events in capital districts (and other more core areas) and a higher proportion of events associated with new and more threatening developments in peripheral areas. The bar for something being reported (esp with geographic precision) is higher in peripheral areas, so there will in peripheral areas be a selection towards more newsworthy events like the appearance of new actors, the territorial advance/retreat of some actor, or the loss of control by the government (or the seeming threat thereof) – all of which may imply higher levels of threat than otherwise equivalent violence and higher levels of threat may generate more displacement.

This might explain or be part of the explanation for the findings in Figure 3d provincial capitals vs non-capitals. It could possibly effect other findings also, like those on IS and on previous violent events, but that is less obvious.

While there is little that can be done about the lack of geographic precision, I think the authors need to take serious potential problems it raises and at least discuss potential implications.

To understand the problem I suggests the authors consult the sources below (they also give references to potential further readings).

On how reporting is related to core-periphery using military SIGACTS data for Afghanistan and comparing it to an earlier version of the UCDP data:

Nils B. Weidmann 2014. "On the accuracy of Media Based Conflict Event Data" *Journal of Conflict Resolution* 59(6): 1129-1149 <https://doi.org/10.1177/0022002714530431>

On how news evaluation works and might affect what gets reported and from where (i.e. what is newsworthy in different contexts):

Galtung, Johan, Ruge, Mari Holmboe. 1965. "The Structure of Foreign News: The Presentation of the Congo, Cuba and Cyprus Crises in Four Norwegian Newspapers." *Journal of Peace Research* 2 (1): 64–91.

Öberg, Magnus, Sollenberg, Margareta. 2011. "Gathering Conflict Information Using News Resources." In *Understanding Peace Research: Methods and Challenges*, edited by Höglund, Kristine, Öberg, Magnus, 47–73. New York: Routledge.

Thank you for this excellent and insightful observation, and for providing very helpful references. We agree that this is an important issue that was not sufficiently discussed in the prior manuscript. The revised paper makes several changes to acknowledge and address this limitation.

First, we now flag very early in the paper (i.e., in the *Introduction*) that there are issues to consider with missing violence data. The relevant text reads, "We use approximately six thousand events in our analysis, corresponding to the subset of all events that are recorded with sufficient spatial and temporal precision (53% of all recorded events); the limitations of this data source are discussed in the *Limitations* section of this paper."

Second, we have created a new “Limitations” section in the paper, and have added discussion of the key points you make above. We thank you again for highlighting the specific core/periphery issue, and for the references -- both of which we have incorporated into this discussion. The relevant text reads, “We also note limitations of the data we use to measure violence and conflict. These data are collected from media sources, supplemented by NGO and IGO reports, field reports, and books (Sundberg and Melander 2013). But not all events are included in media reports; inclusion is affected by a large number of factors (Öberg and Sollenberg 2011). For example, more populous regions and economic centers have a larger media presence and are thus better covered than peripheral areas; contested areas and places with less infrastructure tend to have poorer coverage. Media coverage is also influenced by how “newsworthy” the event is likely to be (Galtung and Ruge 1965). Our analysis is further restricted to those events for which the date and location (i.e., the district) are known. Since the spatial and temporal precision of event reporting is not random (Weidmann 2015), and that non-randomness may even differ across regions of Afghanistan, this could introduce bias into our analysis. For instance, if the threshold for reporting violence in rural areas is higher than the threshold in urban areas (Galtung and Ruge 1965, Öberg and Sollenberg 2011), that could bias our analysis to finding larger effects of violence in rural areas. We perform robustness tests in *Materials and Methods* (Section J) to assess the likely magnitude of this bias, but we also acknowledge that this is a limitation of our data -- and a limitation shared by most empirical work on conflict.”

Finally, to assess the potential biases introduced by our removal of violent events that are not recorded with sufficient (district-level) precision, we performed additional analysis to determine whether the spatial precision with which an event is recorded is correlated with the magnitude of the displacement effect. Specifically, we estimate separate coefficients for each of the three types of events that we use in our analysis, based on their available geographic precision: (A) Events in which the exact location of the event is known and coded (N=1698), (B) Events which occurred within a 25 km radius around a known point (N=789) and (C) Events in which only the district is known (N=969). (This analysis is imperfect, because we would ideally want to know what the response is for events with location not known to a district-level precision (i.e., the events not included in our paper), but of course we cannot do that analysis with the data we have.)

This new analysis is presented in Section J (Implications of missing violence data) of *Materials and Methods*, and Figure S6 presents the results, using an analogous empirical strategy as that used to construct Figures 2a and 3a-d in the manuscript. If it is indeed the case that events recorded with lower precision represent less threatening events, then we might expect coefficients for (C) to have smaller magnitudes than that for (A) and (B).

This does not appear to be the case: there are differences in the point estimates, but the confidence intervals of the estimated coefficients for the three types of events overlap.

We acknowledge that this analysis does not eliminate the concerns you raise above, and it doesn't directly address the specific core/periphery concern. However, it is reassuring that the magnitudes and overall conclusions that we draw in our main analysis are not substantively changed by analyzing these separate groups of events.

The relevant text in Materials and Methods (Section J) reads:

As discussed in Limitations, our analysis does not include violent events that are not associated with specific locations (i.e., where we do not know the district in which the event occurred). This could introduce bias into our analysis if certain types of violence (with specific migration responses) are systematically more or less likely to have known locations. We therefore conduct additional analysis to determine whether the spatial precision with which an event is recorded is correlated with the magnitude of the displacement effect.

Specifically, using the same empirical approach described in Section G, we create separate treatment indicators for each of the three types of events that we use in our analysis, based on their available geographic precision: (A) Events in which the exact location of the event is known and coded, $A_{\{d,t+\tau\}}$ (N=1698), (B) Events which occurred within a 25 km radius around a known point, $B_{\{d,t+\tau\}}$ (N=789), and (C) Events in which only the district is known, $C_{\{d,t+\tau\}}$ (N=969). These replace the treatment indicators in (1):

$$\begin{aligned}
 & g(\mathbb{E}(Y_{dt,k} | X_{dt}, A_{d,t+\tau}, B_{d,t+\tau}, C_{d,t+\tau})) \\
 & = \gamma_d + \lambda_t + \sum_{\tau=-30}^{180} \beta_{A,\tau} A_{d,t+\tau} + \sum_{\tau=-30}^{180} \beta_{B,\tau} B_{d,t+\tau} + \sum_{\tau=-30}^{180} \beta_{C,\tau} C_{d,t+\tau}
 \end{aligned} \tag{5}$$

The results, shown in *SI Appendix Figure S7*, indicate that the displacement response is very similar for violent events with these three different levels of spatial precision. There are small differences in the point estimates but the general pattern of the response is unchanged, and the confidence intervals of all three violence types overlap. Of course, this analysis does not eliminate the possibility that there might be a qualitatively different displacement response to violent events that are not recorded in our dataset (or for which district information is unknown). Unfortunately, we cannot directly test that concern, since we cannot estimate the displacement effect of violence when the location of the violence is not known.

Comment 2:

Second, scope conditions. How far do the findings in this manuscript travel? Are the findings particular to, or contingent on some aspect of this context, or can we expect similar patterns to obtain elsewhere? As a conflict country Afghanistan may in some ways be an outlier.

Afghanistan have a 40+ year history of war and displacement. Millions of people have left, and it seems likely that 40 years of fighting has led to some social adaptations and strategies for coping with violence and threats of violence that are not present in other contexts. They have several ethnic and sectarian divisions layered on top of a partly disrupted clan based society all of which is reflected in the armed groups. And there is much foreign involvement and interference. So, for example, does the tendency to flee to district capitals have something to do with clan and ethnic or sectarian divisions (where one can safely flee), or is it because of safety in numbers, or is it because of stronger

government control/presence in the district capitals as the authors suggest? Presumably the idp's seek some measure of safety, but government control does not always imply safety, even if it means less fighting. Does the age structure affect the propensity to flee? The median age in Afghanistan is just over 18 years and it is one of the poorest countries on the planet. Would patterns be similar in a country with a different age structure and more resources? These are not the only questions and I do not know if they are necessarily the most salient ones so I do not expect answers to them specifically. But I think some mention or discussion of what the authors think about the scope conditions of their findings are in order. It is valuable for future research. It is also, I think, important for ethical reasons to at least flag clearly if the authors are not able to delineate the scope conditions so that policy makers and others who may act of these findings in other settings are

aware that they may not apply - at least not equally - in other cases.

Thank you for these questions and suggestions. We agree that this is an important and useful discussion, and have included a paragraph in *Limitations*. Briefly, we believe the general result that displacement is affected by violence is likely to be pervasive, but specific results (including magnitudes and elements of heterogeneity) may not generalize, due to some of the factors that you rightfully suggest. Some specific examples that we cite include Afghanistan's long history of conflict resulting in different adaptations, and location-specific results that are potentially influenced by many context-specific factors. We have taken the reviewer's suggestion to urge policy- and other decision- makers to be cautious when interpreting and extrapolating from our findings. The relevant text reads, "However, the Afghan setting is unique in many respects, and some results (such as the magnitude of the estimated effect, or specific elements of the heterogeneous response) may be specific to Afghanistan. For example, Afghanistan has a decades-long history of conflict; our results will likely translate better to settings with endemic conflict than ones where violence is new. Afghanistan also has strong clan-based, ethnic, and sectarian divisions; a strong patriarchal society; and a relatively poor and young population. These contextual factors likely impact migration decisions in complex ways; our results will naturally be most relevant to environments with similar characteristics."

Comment 3:

Finally, the subject of this study is the displacement of Afghan people in Afghanistan in the internationalized but still very much Afghan civil war. The war in Afghanistan pre-dates the current US intervention by a quarter of a century or so. Yet, in the abstract a justification for looking at Afghanistan seems to be that it is the longest war in US history. While this may be true, it strikes me as irrelevant to the study and bizarrely US-centric. I would suggest the authors simply drop that part.

Thank you for pointing this out. We agree and have replaced this text. The sentence now reads, "We develop an approach to measure the impact of violence on internal displacement using anonymized high-frequency mobile phone data. We use this approach to quantify the short- and long-term impact of violence on internal displacement in Afghanistan, a country that has experienced decades of conflict."

R3's comments are in black/Arial; our point-by-point responses are in blue/Times

Thank you very much for your careful and expert review of our manuscript, and for the detailed and constructive suggestions. You will find point-by-point response to each of

your Suggested Improvements below.

You may also notice other changes to the manuscript, some of which were done in response to comments from the editor and other reviewers, and others were implemented to meet strict formatting requirements of the journal (including changes to the title and abstract, and the removal of footnotes and italics). We think the end result is a much-improved manuscript, though at times we struggled to balance the desire to thoroughly address reviewers' comments with the space constraints of the journal.

Thanks again for taking the time to review this manuscript!

Comment 1:

In this paper, the authors use mobile phone traces to measure internal displacement within Afghanistan in response to violent events. The authors are able to quantify where people are displaced to and for how long they remain displaced with respect to their home location. All locations are defined to district (admin 2) level. The authors report that there is a causal and significant increase in out-displacement following (and to a lesser extent preceding) a violent event. The nature of the displacement is affected by the group responsible for the attack, the severity and frequency of the violence as well as the proximity to urban/rural areas.

I believe that this paper is a notable and high quality contribution to the scientific literature on conflict, migration and the use of non-traditional data sources for official statistics. I would also add that this work is not a pure academic work but closer to 'scholar-practice' and so it should be reviewed with that in mind.

What this means concretely is the following; a large part of the contribution here is not concerned with methodological advancement but with the enormous work that goes into a complex partnership with a mobile phone operator in a low income country. To navigate the regulations of accessing this data in a responsible way, dealing with data cleaning and bringing to publication the study of such a sensitive topic is quite an achievement. Part of the value is the precedent that this sets for further similar studies and opportunities for other researchers and partnerships and hopefully, eventually, the operationalisation of this kind of work into sustainable partnerships between governmental organisations and mobile phone operators.

The flipside of this is that this study will surely be read by many technically literate and skeptical non-academics who work in National Statistical Agencies and similar. As a result, speaking from deep experience, any far reaching claims made by the authors could potentially be damaging to this endeavour in the long term.

With this in mind, my main substantive request to the authors is to be more clear about the limitations of this work. This is not a criticism of the work itself, it is simply the nature of the events and the data.

We thank the reviewer for the kind remarks. We have reorganized the *Discussion* section to include a specific *Limitations* section, and expanded our discussion significantly based on feedback from the reviewers.

Comment 2:

-- Figure 1 is the most critical of the entire paper: the spatial distribution of cell towers and violent events. Anyone who works with mobile phone data knows that cell towers are heterogeneously distributed and it will surprise no-one that there are parts of Afghanistan with no cell phone coverage. So please break out this figure into 3 parallel maps showing (i) cell tower locations (ii) violent events and (iii) district population so that these can be easily compared.

Thank you for this suggestion. We have included the requested figure as Supplementary Figure 1. In the main text, we reference this figure together with Figure 1.

Comment 3:

-- It is common practice to evaluate the user base in mobile phone studies by a simple comparison of district population from official statistics and mobile subscriber home location. Of course the best official statistics in Afghanistan will be imperfect but this kind of validation is important and will increase the impact of the work in the long term.

Thank you for this suggestion. We performed the suggested analysis by comparing district population as estimated by Afghanistan's Central Statistical Office from 2015-2016 to the mean number of subscribers having home locations assigned to each district, averaged over the same one-year period. The resulting scatterplot is shown in the figure below; the Pearson's correlation coefficient is .94.

We now reference this new analysis in *Materials and Methods* Section C: “In Afghanistan, we find that the geographic distribution of daily modal locations of mobile phone subscribers broadly reflects the geographic distribution of the population. In particular, when comparing the number of mobile phone users in each district to the district population as estimated by Afghanistan's Central Statistical Office, the Pearson's correlation coefficient is 0.94. (We calculate the number of subscribers in each district as the number of subscribers whose daily modal location is assigned to that district, averaged over all days in which the district has a non-zero number of subscribers, for a one-year period. Official estimates are obtained from <https://data.humdata.org/dataset/estimated-population-of-afghanistan-2015-2016>. Pearson's correlation is computed on a log scale.)” To keep the manuscript short, we have not currently included the figure in the manuscript, but we are happy to do so if the reviewer thinks this is important.

Comment 4:

-- Again, common practice in mobile phone studies is to construct a Voronoi tessellation of cell towers and to use the overlap of the Voronoi cells with the administrative districts to, perhaps probabilistically, assign a home location. It's not clear to me from the provided details and citation (A general approach to detecting migration events in digital trace data) what the authors did instead and why. Some more explanation here would be welcomed.

We apologize for the lack of clarity in how we assign home locations. We have revised and expanded the relevant discussion in *Materials and Methods* Section C (“Measuring

migration”) to provide more detail. This section now reads:

Our first step is to derive a ‘daily modal location’ for each subscriber for each day, which is intended to capture the district in which the subscriber spends the majority of their time on that day. For each individual, we first compute their most commonly used cell tower in each hour. Then, for each 24-hour period from 6 A.M. to 6 A.M. the next day, we compute the mode of the hourly modal towers. The towers are then mapped to districts using point-in-polygon assignment. Similar methods have been used and validated in other work (Phithakkitnukoon et al. 2011, Toomet et al. 2015, Hong et al. 2019, Hankaew et al. 2019, Deville et al. 2014, Vanhoof et al. 2018). While several prior studies use night-time hours to infer daily locations; we instead use all hours, which allows us to include more individuals in our analysis. For example, in April 2013, data are available for approximately 31 million individual-days using night-time hours (6 P.M. to 6 A.M.), while 61 million individual-days are defined when using all hours (6 A.M. to 6 A.M.). The two approaches are highly correlated: Of the 31 million observations available using night-time hours, 89% record the same daily modal districts when computed using all hours. Another common approach in the literature divides the physical terrain into approximate catchment areas of each cell tower, using a Voronoi tessellation (e.g., Gonzalez et al. 2008, Deville et al. 2014). Our analysis focuses on slightly larger administrative districts, since many of our violent events are only identified at the district level.

In Afghanistan, we find that the geographic distribution of daily modal locations of mobile phone subscribers broadly reflects the geographic distribution of the population. In particular, when comparing the number of mobile phone users in each district to the district population as estimated by Afghanistan's Central Statistical Office, the Pearson's correlation coefficient is 0.94. (We calculate the number of subscribers in each district as the number of subscribers whose daily modal location is assigned to that district, averaged over all days in which the district has a non-zero number of subscribers, for a one-year period. Official estimates are obtained from <https://data.humdata.org/dataset/estimated-population-of-afghanistan-2015-2016>. Pearson's correlation is computed on a log scale.)

These daily modal locations tend to be sparse and noisy: for instance, many people do not use their phones on every single day; people may take short trips to nearby (non-residential) locations, and so forth. Thus, our second step employs an unsupervised scanning algorithm (Chi et al. 2020) to identify contiguous segments in which a subscriber is, with high probability, resident in a single district. This algorithm helps smooth the influence of noise (for instance, long periods where a person is primarily in one location, but intermittently visits other

locations for one or two days) and missing data (for instance, when a person uses their phone infrequently, but almost exclusively from the same location), and has the advantage of not arbitrarily grouping days into calendar weeks or calendar months.

You also raise the valid point that an alternative approach to inferring locations would probabilistically assign people, based on their cell tower, to administrative districts. Note that our current point-in-polygon approach is almost identical to deterministically assigning people, based on their cell tower, to the administrative district with which that tower has the most overlap (it will not be identical if districts are not convex, but in practice this is rare). We do not take the probabilistic approach because it would significantly complicate our panel event study estimation framework, which requires that each individual be associated with a single district on each day. There are likely alternative modeling approaches that would not have this limitation, but we are reluctant to change our workhorse model at this stage in our project.

Comment 5:

-- An overarching issue in studies of conflict and other extreme events is that they are very sparse. The case of Afghanistan is no different, by my estimate there are roughly $365 \times 4 \times 398 \sim 600k$ district-days in the data, with $\sim 3k$ violent events with district information giving a frequency of $\sim 0.5\%$. This may have some consequences for the validity of the causal analysis, I'm not qualified to comment on that, however at the very least the authors should make some reference to this aspect of the data and what confounding effects it might introduce.

Thank you for raising this concern. In the manuscript, we now explicitly note the sparsity of the violence data. We also clarify that the statistical model we employ is robust to sparsity; as long as events are recorded accurately, estimates will not be biased because of this. The relevant text reads, "although the treatments (violent events) occur relatively infrequently, the statistical model we employ is robust to sparsity; if all of the events are recorded accurately, estimates will not be biased because of sparsity."

However, if the sparsity is not random, and if specific types of events are systematically missing, that could bias our estimates -- this is the focus of your next comment, so we will respond to that concern below. We also state this in the manuscript, "Non-random missingness of recorded events could bias estimates and are discussed in *Limitations*."

Comment 6:

-- Relatedly, the authors report that only 47% of the ground truth data on violent events have district level spatial accuracy. Again, this is no fault of the authors but should be addressed

clearly. In fact, the use of passively collected data for proxying measures of interest is increasingly held back by a lack of high quality data against which to validate. The authors should comment on (i) what the elimination of roughly half the violent events might mean for their results and (ii) how these shortcomings in conflict data might be addressed in the future. I can recommend the following paper as an entry point into this debate; Growing pains for global monitoring of societal events (2016), Wang et al (<https://science.sciencemag.org/content/353/6307/1502.summary>)

Thank you for pointing this out and for the suggested reference. We agree that this is an important issue. The revised manuscript is now more explicit about this limitation. For instance, the introduction now notes, “We use approximately six thousand events in our analysis, corresponding to the subset of all events that are recorded with sufficient spatial and temporal precision (53% of all recorded events); the limitations of this data source are discussed in the *Limitations* section of this paper.”

We have also added a new section on “Limitations”, which includes a discussion of this issue: “We also note limitations of the data we use to measure violence and conflict. These data are collected from media sources, supplemented by NGO and IGO reports, field reports, and books (Sundberg and Melander 2013). But not all events are included in media reports; inclusion is affected by a large number of factors (Öberg and Sollenberg 2011). For example, more populous regions and economic centers have a larger media presence and are thus better covered than peripheral areas; contested areas and places with less infrastructure tend to have poorer coverage. Media coverage is also influenced by how “newsworthy” the event is likely to be (Galtung and Ruge 1965). Our analysis is further restricted to those events for which the date and location (i.e., the district) are known. Since the spatial and temporal precision of event reporting is not random (Weidmann 2015), and that non-randomness may even differ across regions of Afghanistan, this could introduce bias into our analysis. For instance, if the threshold for reporting violence in rural areas is higher than the threshold in urban areas (Galtung and Ruge 1965, Öberg and Sollenberg 2011), that could bias our analysis to finding larger effects of violence in rural areas. We perform robustness tests in *Materials and Methods* (Section J) to assess the likely magnitude of this bias, but we also acknowledge that this is a limitation of our data -- and a limitation shared by most empirical work on conflict.”

The above text references a new section, “Implications of missing violence data”, that provides additional empirical analysis. The main result of that analysis is that we do not see a correlation between the magnitude of the migration response and the precision of spatial information for the violent event. We take this as encouraging -- but not conclusive -- evidence that heterogeneous spatial accuracy is not heavily influencing our results.

Finally, we have added a brief discussion of what we hope for the future for conflict events data. The relevant text reads, “While these problems of exclusion are faced by all

violent events data sets that rely on news sources, and there is little we can do to improve the quality of the data, we are optimistic that the quality of conflict event data will improve as they gain more widespread use. For instance, several papers now articulate best practices for the collection of conflict data (Salehyan 2015, Wang et al. 2016), and researchers are increasingly exploring how social media and other non-traditional data can be used to source conflict data (Zeitsoff 2011).”

Comment 7:

Finally, I would be interested in the authors perspective on the degree to which this is a descriptive analysis vs a predictive endeavour. I would imagine that there is a delay in reporting and compiling violent events in the Uppsala database and similar. Would it be possible to use mobile phone derived measures of displacement as an early warning of a violent event (particularly in light of the previous points regarding sparsity and false positives)?

This is an interesting point that we considered early on in this project. In principle, we agree that this might be feasible, and we are aware of one recent paper that explores this in detail (Berger et al. 2014). However, as you might imagine, this is a complex and subtle problem, and to properly address it would require serious investigation. For this reason, we decided it is out of scope for the current paper, although we now mention this in the discussion as an interesting dimension: “Our analysis also indicates that people appear to anticipate the occurrence of violence , leaving before it occurs --- a finding that relates to prior work on the predictability of violence and conflict (Ward et al. 2010, Hegre et al. 2013, Cederman and Weidmann 2017, Bazzi et al. 2021, Kapoor and Narayanan 2021), including recent work using mobile phone data (Berger et al. 2014).”

Decision Letter, first revision:

19th August 2021

Dear Josh,

RE: "Mobile Phone Data Reveal the Effects of Violence on Internal Displacement"

Thank you for submitting your revised manuscript and for all your work on the revision.

Although your manuscript has been revised in response to reviewer comments, it does not fully comply with our editorial policies and formatting requirements. In particular, we require that inferential statistical results be fully reported, including coefficients/effect sizes, p-values, and confidence intervals. Additionally, all main comparisons/statements of differences should be supported by formal inferential tests. For instance, the comparison between the Taliban and IS in the paragraph beginning line 82 should include a formal test of the difference between the two groups.

Before we can send the manuscript back to our reviewers, we ask that you revise it to ensure that it complies fully with our policies by including p-values in addition to confidence intervals and adding statistical tests of differences as needed. To assist with this process, I have attached another copy of our checklist. If you are uncertain as to how to address any of these points in the checklist or have any questions, please don't hesitate to contact me.

[REDACTED]

Thank you in advance for attending to these requests and I look forward to receiving your revised manuscript.

Sincerely,
Aisha

Aisha Bradshaw, PhD
Senior Editor
Nature Human Behaviour

Author Rebuttal, first revision:

R1's comments are in black/Arial; our point-by-point responses are in blue/Times

Thank you for your extremely thoughtful and constructive review, and for the generally enthusiastic assessment of our manuscript. We have provided a point-by-point response to each of your Suggested Improvements below.

You may also notice other changes to the manuscript, some of which were done in response to comments from the editor and other reviewers, and others were implemented to meet strict formatting requirements of the journal (including changes to the title and abstract, and the removal of footnotes and italics). We think the end result is a much-improved manuscript, though at times we struggled to balance the desire to thoroughly address reviewers' comments with the space constraints of the journal.

Thanks again for taking the time to review this manuscript!

Comment 1:

Given that mobile phone data is a measure of communication by definition, I'm very curious about whether the authors can show any information transmission mechanisms for their argument. Do call volumes change before or after violence? If a given violent event is motivating migration, I would expect call volumes to rise immediately after that violent event as people discuss with each other what happened and what to do about it. That could also allow the authors to specify how exactly civilians are anticipating violence.

This is an interesting point. We have conducted basic analysis that suggests that there is, on average, a spike in call volumes on the day of violence. This is consistent with your suggestion that call volume rises immediately after violence as people discuss what

happened and what to do. The figure below shows the results of a regression analogous to what we use in the manuscript (Equation 1). The response variable is the number of calls recorded per district day, and we use leads and lags of up to 180 days.⁵ The interpretation of each plotted coefficient is “the multiplicative increase in number of calls made in the district, on average, when violence occurs in the district at lag t , holding all other variables constant.” The coefficient significantly larger than 1 on the day of the violence indicates that call volume is more than 2.5% higher, on average, on days in which violence occurs, compared to non-violent days.

This line of analysis is something that has been explored in a few papers that we are aware of, such as Bagrow et al. 2011 and Gao et al. 2014. After careful consideration, we elected to not incorporate this new analysis into the manuscript, as we were concerned that it might distract from our main focus, which is the effect on out-migration and displacement. However, we now briefly note that this response exists, and point the reader to prior work that explores the question in greater depth. The relevant text reads, “Rumors and word-of-mouth may also play a role: prior work suggests that information about violence spreads quickly (Bagrow et al. 2011, Gao et al. 2014) -- including unverified rumors (Silverman et al. 2021) -- and that rumors may prompt people to take action to protect themselves (Schon 2021). Consistent with this idea, we do observe a significant increase in call volume on days on which violence occurs.”

(We revisit the issue of rumors in response to your comment #9 below.)

Comment 2:

Since violence is spatially and temporally autocorrelated, how can the authors specify that migration is responding to a specific violent event? Where they observe migration as an

⁵ We use a log transformation on the response variable (the number of calls is extremely right-skewed), fit the model using ordinary least squares, with standard errors clustered at a district level. We include district-days with one or more calls.

anticipation of violence, how can they tell that civilians are not instead reacting to previous violent events? It appears to be a serious limitation that the authors cannot distinguish responses to specific events vs. responses to a broader context of violence or dangerous shifts in conflict dynamics. I'm not sure if the authors can resolve this, so if they can't then they need to explicitly mention this limitation in the text. This challenge is an issue that Schon (2015) and Schon (2019), which the authors cite, acknowledge and grapple with.

Thank you for raising this important point. In the revised manuscript, we are now much more explicit about what concerns our modeling strategy allows us to address, and what limitations still remain.

Specifically, we use a panel event study model, which takes into account spatially and temporally autocorrelated violence by including leads and lags for violence indicators (up to 180 days in the past and 30 days into the future). Thus, the analysis is essentially asking, “On average, are more people leaving a particular district on a particular day that violence occurs *relative to the number who leave that district on a non-violent day, controlling for other violence that occurs in the 180 days before and 30 days after?*” In this way, we are estimating the idiosyncratic increase in out-migration associated with each specific violent event, averaged across all violent events. It may be the case that one single event was preceded by several violent events, and so for that event our estimate of “day of” displacement could capture some of the displacement from the prior events. But after incorporating displacement data covering all violent events in the country, our analysis tells us what the “average” displacement effect of violence is, on the day of violent events, as well as in the 180 days prior and 30 days post.

We have reorganized the paper to try to clarify these points. Specifically, the *Introduction* now provides more details on our modeling strategy: “Finally, we use a high-frequency panel event study design to estimate the causal effect of violence on migration, which we measure as the increase in movement of subscribers out of a district impacted by violence, relative to movement from the same district on non-violent days, while controlling for seasonality and other temporal factors. Specifically, we regress total out-migration from each district on each day on a vector of binary indicators for whether violence occurred in that district on that day, in the 180 days prior (the ‘lags’) or the 30 days after (the ‘leads’) that day. We control for district and day fixed effects, and cluster standard errors at the district level. This approach allows us to estimate the “average displacement effect” of violence -- averaged over the 5,984 violent events in our dataset - - relative to displacement that occurs in the absence of violence. This modeling approach and its identifying assumptions are described in detail in *Materials and Methods*, Section E.”

In addition, we have added a new subsection, “Summary of Identification Strategy”, to *Materials and Methods* Section E, which is our attempt to succinctly describe what our model is able to estimate. The text of this section is reproduced below:

To more plainly summarize our statistical approach to measuring the effect of violence on displacement: We regress out-migration on each district-day on indicators for occurrences of violence up to 180 days prior, and up to 30 days in the future, while controlling for geographic and temporal factors. This approach is designed to capture out-migration in excess of the out-migration that normally occurs in that district (on all other days) and on that day (in all other districts). Thus, the model does not assume that people do not move when violence is not occurring; instead, it uses movement in non-violent times and places as a baseline, to better isolate the *additional* movement that co-occurs with violence.

We include 180 lag terms and 30 lead terms in order to measure excess out-migration (again relative to normal out-migration) that occurs in the 180 days after violence, and that occurs in the 30 days leading up to violence, as well as excess out-migration on the day of violence. Since violent events may be spatially and temporally correlated, a single observation (district-day) in the regression could have multiple violence indicators that are turned on; the migration dependent variable for that observation would thus contribute to the estimation of violence effects on all of the affected leads and lags.

Using this regression framework allows us to estimate the "average displacement effect" of violence, averaged over the 5,984 violent events in our dataset. For example, a coefficient of .03 on the indicator for violence at a lag of 10 days can be interpreted as "on average, violence occurring 10 days prior increases the odds of migration out of a district by 3% (a multiplicative change of $e^{.03} \sim 1.03$) holding all other variables constant." This approach helps limit the extent to which any one specific event, which might have unusual characteristics or correlates, can influence our final results. For instance, if one violent event happened to occur on a day in which a certain district would have seen unusual out-migration even in the absence of violence, that single event would have limited impact on our final estimates. The main concern is if violent events were systematically correlated with other unobserved factors -- above and beyond the flexible spatial and temporal fixed effects that we control for in the regression.

More broadly, however, it is possible that unobserved factors in the environment (such as tension, or even violence that occurs but does not result in a fatality) could precede or co-occur with the violent events that we do observe. In this case, we would be mis-attributing the displacement to the observed violent events, rather than the unobserved contextual factors. In addition to the above text (and the "main concern" noted in the last sentence quoted above), we now acknowledge this issue more explicitly in the *Limitations* section as follows:

A broader limitation of our quantitative approach is that we cannot say much about what specific aspects of violence people are reacting to, and whether

observed displacement is voluntary or not. In aggregating effects over all violent events, we find that, on average, violence causes displacement -- but this may hide the fact that certain types of violence lead to a reduction in mobility (for instance, if populations are trapped or involuntarily immobile (Lubkemann 2008)), if that reduction is offset by other types of violence that dramatically increase out-migration. Related, in discussing the anticipatory response, we noted that recorded (fatal) events may be preceded by a broader context of non-fatal violence, and individuals may in fact be responding to broader changes in the local environment that we cannot directly observe in our data.

Comment 3:

I would like to see at least a brief mention of whether there are any concerns about cell towers getting destroyed or damaged during violent periods. Could it be a threat to measurement if armed groups damage cell towers, thereby blocking cell phone signals and causing a notable drop in calls that can be completed? The issue of conflict destroying infrastructure is discussed in a wide variety of research, such as Schultz & Mankin (2019) on how weather stations get damaged or destroyed in African conflict zones.

Thank you for raising this interesting point. We have conducted additional analysis to explore this in more detail. This analysis suggests that (i) there is, on average, a small decrease in the number of active cell towers in the period surrounding violence; but (ii) in general, single tower outages are not severe enough to reduce coverage at the district level.

Specifically, we include two figures below, both of which are generated using regressions analogous to what we use to report our main results in the manuscript (Equation 1). In the first figure, the response variable is the number of active towers in a district on a given day.⁶ The second figure uses a binary response variable that indicates whether a district has at least one active tower each day.⁷ Both figures include leads and lags of up to 180 days.

⁶ We fit the model using ordinary least squares, with standard errors clustered at a district level. District-days in which there are 1 or more active cell towers are included.

⁷ Here we include all district-days for districts with at least one active cell tower during our measurement period (2013-2017). We fit a linear probability model using ordinary least squares, with standard errors clustered at a district level.

The first figure indicates that, starting roughly two months before fatal violence, there is, on average, a slight decrease in the number of active towers. This suggests that cell towers might be destroyed or turned off prior to recorded fatal events. News reports have suggested that a tactic the Taliban employs to restrict communications is to ask mobile phone operators to turn off coverage, or destroy towers, which is consistent with these results.⁸ There are several possible implications of this reduction in active cell towers: (1) individuals connect to a different cell tower within the same district, (2) individuals connect to a cell tower in a different district, and (3) individuals are unable to get service. How would these impact our analysis? If it is just (1), that should not affect our results, since all of our analysis is done at the district level. (2) is more of a concern, since it could upward bias our estimates of out-migration. However, Afghan districts are quite large, and towers are frequently redundant within districts (particularly in more populous

8

<https://www.voanews.com/extremism-watch/taliban-warns-phone-companies-shut-down-their-coverage-ghazni>, <https://www.nytimes.com/2019/10/02/world/asia/afghanistan-taliban-cell-towers.html>

areas). And as can be seen in the second figure, violence does not significantly impact whether the district as a whole has one or more active towers. (Also note that if *all* individuals are only able to connect to towers in a different district, those observations get discarded, and would not influence our results.) Finally, for (3), when individuals are able to reconnect to a cell tower within the district, they will be recorded after they are able to reconnect, resulting in a period with no data. In our data processing (described in *Materials and Methods* Section C), sparse segments in which individuals have no sustained activity in a different district are interpolated, so they will be recorded in the same district, and this is unlikely to bias results.

In our revised manuscript, we now provide a brief summary of this analysis, which reads, “News reports have described the Taliban restricting access to communications or destroying cell towers, and we do see a small reduction in the number of active cell towers in a district during periods of violence. However, we do not see significant decreases in call volumes at a district level, nor do we see a decrease in the probability of a district having an active tower, likely indicating that individuals are able to connect to a different cell tower within the same district. If it is the case that all individuals are only able to connect to a cell tower in a different district, our response variable would be a 1 and hence dropped from the regression. This limits overestimation of the displacement response.”

Comment 4:

By using mobile phone data, the authors are measuring the mobility of people. It is unclear which movements are displacement and which movements are part of normal mobility patterns. This is the biggest challenge with the data. Not all movement is displacement, and the authors do not have pre-conflict data to use as a source of comparison to help them distinguish displacement from mobility. The authors need to do more to engage with this challenge. It looks like they are assuming that mobility patterns that suddenly change are displacement, but this assumption likely over-classifies movement as displacement. Is there some way for the authors to identify what a “normal mobility pattern” looks like? Then, the authors could describe how a different mobility pattern can be interpreted as displacement.

We apologize for the lack of clarity in how we previously described our modeling approach. Your comments have helped us understand how confusing that description was, and (we hope) have enabled us to write a more complete and intuitive version. In the text quoted above (in our response to Comment #2; from *Materials and Methods* Section E), we have tried to address this concern. As noted there, it is not just sudden changes in mobility that we count as displacement, but rather changes in mobility in excess of the “normal mobility pattern” that you describe. The model defines “normal” as the mobility typical of each district (across the entire period for which we have mobility data), after accounting for what is typical on each day (across all districts in Afghanistan).

Comment 5:

The approach that the authors use is an interesting attempt to get at this idea. Yet, their approach assumes away the possibility that people will move when violence is not occurring. This is a potentially serious problem, since there is research showing that violence can deter movement (see work on involuntary immobility or trapped populations). Civilians may wait until the real or proverbial smoke clears before they move. I'm willing to be convinced that this challenge cannot be fully solved, but the authors would then need to discuss in the main text that future research could build on this by explicitly measuring internal displacement immediately following violence and internal displacement that occurs after some delay.

We believe this comment follows directly from Comments #2 and #4, and again results from our lack of clarity in describing our modeling strategy. We hope the additional explanations, discussed above, help address this concern. But just to restate: we do not assume that people do not move when violence is not occurring. Rather, the statistical model, by way of the district fixed effect, uses the movement out of each district on all days in the dataset (the vast majority of which are non-violent) as a baseline to determine the excess (unusual) movement that occurs on days with violence.

You also raise the valid point that violence can deter movement, and cause involuntary immobility. It is certainly possible that this occurs in Afghanistan, though unfortunately the data we have do not make it easy to investigate this hypothesis. In essence, our analysis says that, on average, out-migration increases in times of violence. However, it is possible that some violence deters migration, but those events are more than offset by other violence that increases migration. In the revision, we more clearly acknowledge this limitation in the new *Limitations* section, the relevant text of which is reproduced here: "A broader limitation of our quantitative approach is that we cannot say much about what specific aspects of violence people are reacting to, and whether observed displacement is voluntary or not. In aggregating effects over all violent events, we find that, on average, violence causes displacement -- but this may hide the fact that certain types of violence lead to a reduction in mobility (for instance, if populations are trapped or involuntarily

immobile (Lubkemann 2008)), if that reduction is offset by other types of violence that dramatically increase out-migration.”

Comment 6:

The analysis would benefit from considering the role of territorial control. There is available data on whether the government or the Taliban controls a piece of territory from the Special Investigator General for Afghanistan Reconstruction in their quarterly reports. It is not a perfect resource, but it can give the authors an indication of whether violent events are occurring in government-controlled territory. The consequences of violent events are likely to be different depending on which armed group controls that territory.

Thank you for this excellent suggestion. We have looked into the SIGAR reports, and our understanding is that SIGAR began assessing territorial control at a district level in December 2015. Unfortunately, the quarterly reports that we have been able to obtain only include a summary and not a classification of individual districts, for example: ‘USFOR-A reports that approximately 70.5% of the country’s districts were under Afghan government control or influence as of January 29, 2016. Of the 407 districts within the 34 provinces, 287 districts were under government control or influence, 26 districts (6.4%) within 11 provinces were under insurgent control or influence, and 94 districts (23.1%) were “at risk.”’ The only available map and list of districts we have found is from October 2017, and was provided as an addendum to the 2018 Q1 report (https://www.sigar.mil/pdf/quarterlyreports/Addendum_2018-01-30qr.pdf). These data are not ideal, as territorial control changes over time, but we nonetheless use them to assess whether districts that were under Government vs. Taliban control in 2017 have different responses to violence. Empirically, we use a regression specification similar to the one used to compare provincial capitals vs. non-capitals (see *Materials and Methods* Section G).

The figure below shows the results for N=1469 events that occurred in districts under government control or influence vs. N=1885 events in districts that are contested or under insurgent control or influence (i.e., districts where SIGAR indicates insurgent control, insurgent influence, or contested control). We see a larger displacement response in districts under Taliban control/contested.

In the manuscript, we briefly describe these results in the main text as follows, “We also find suggestive evidence that displacement effects are highest in areas that are under insurgent control, or contested by the Taliban (*SI Appendix Figure S2*). We do not emphasize these results, since our data on insurgent control are not ideally suited for this analysis (see *Limitations*).” Note that we do not feature these results more prominently because we have concerns about the underlying data, which we discuss in the *Limitations* section as follows:

Our analysis is further constrained by the general scarcity of publicly available quantitative data covering Afghanistan during this period. For instance, *SI Appendix Figure S2* provides suggestive evidence that displacement effects are greatest when violence occurs in areas controlled or contested by the Taliban -- a finding that is supported by prior work on the dynamics of civil wars (Esparza et al. 2020). However, we consider such results preliminary, because they are based on a cross-sectional assessment of territorial control conducted in October 2017. Since territorial control changed rapidly during the period we study, a more reliable analysis would require panel data on territorial control over time.

Comment 7:

The authors should consider involuntary immobility, and how that may threaten their findings. In this respect, I suggest the following articles as useful to consider:

Lubkemann, S. (2008). "Involuntary Immobility: On a theoretical invisibility in forced migration studies." *Journal of Refugee Studies* 21(4): 454-475.

Schon, J. (2016). "The centrality of checkpoints for civilians during conflict." *Civil Wars* 18(3): 281-310.

Thank you for pointing this out. We have included this in the *Limitations* section -- we are unable to distinguish involuntary immobility from voluntary immobility, any

involuntary immobility will result in an underestimation of the effect of violence on displacement.

Comment 8:

When discussing the ability of civilians to anticipate violence, the authors may benefit from the following:

Rubin, J. and W. Moore (2007). "Risk Factors for Forced Migrant Flight." *Conflict Management and Peace Science* 24(2): 85-104.

Uzonyi, G. (2014). "Unpacking the Effects of Genocide and Politicide on Forced Migration." *Conflict Management and Peace Science* 31(3): 225-243.

Thank you for these suggestions. We have expanded the discussion of anticipatory effects, and included one of the suggested references.

Comment 9:

Specifically, when discussing civilian threat perceptions, the authors may benefit from acknowledging the role of rumors in conflict zones:

Allport, G. and L. Postman (1947). *The Psychology of Rumor*. New York, Russell & Russell.

Greenhill, K. M. and B. Oppenheim (2017). "Rumor has it: The adoption of unverified information in conflict zones." *International Studies Quarterly*: sqx015.

Larson, J. M. and J. I. Lewis (2018). "Rumors, Kinship Networks, and Rebel Group Formation." *International Organization*: 1-33.

Schon, J. (2020). "How narratives and evidence influence rumor belief in conflict zones: Evidence from Syria." *Perspectives on Politics*: 1-14.

Silverman, D., Kaltenthaler, K., & Dagher, M. (2021). *Seeing Is Disbelieving: The Depths and Limits of Factual Misinformation in War*. *International Studies Quarterly*.

Thank you for these suggestions. The revised manuscript now contains a discussion of the potential role of rumors, which reads, "Rumors and word-of-mouth may also play a role: prior work suggests that information about violence spreads quickly (Bagrow et al. 2011, Gao et al. 2014) -- including unverified rumors (Silverman et al. 2021) -- and that rumors may prompt people to take action to protect themselves (Schon 2021). Consistent with this idea, we do observe a significant increase in call volume on days on which violence occurs."

Comment 10:

The abstract is clear. There are just unnecessary commas after the word events and before the word cause. My only comment here is that the authors may want a plainer way to call provincial capitals “resilient to local violence.” Resilience has a ton of meanings across academic disciplines and topics, so I think it would be better to use different language there.

Thank you for this suggestion. We have removed this part of the sentence (due to word limits on the abstract), which now reads, “Provincial capitals act as magnets for people fleeing violence in outlying areas.”

Comment 11:

In the opening paragraph of the Introduction, the estimate that two-thirds of the global extreme poor will be living in conflict and fragile situations is a bit odd. I don't think it helps the authors. The first half of this sentence is fine, but the prediction by 2030 implies a lot of things about the future of conflict that I don't think the authors want to get into. There is no cost to them of just dropping this.

Thank you for pointing this out. We have removed the second half of that sentence.

Comment 12:

I'm not sure if the journal has requirements influencing this, but I found the sections out of order for how I read manuscripts. I would prefer to see the Materials and Methods discussion come before Results and Discussion. This is a simple switching of the order of sections, so it is an easy edit.

Thank you for this suggestion -- we are unable to make this change due to the journal's formatting requirements.

R2's comments are in black/Arial; our point-by-point responses are in blue/Times

Thank you very much for your positive assessment of our manuscript, and for providing very clear and constructive feedback. You will find point-by-point response to each of your Suggested Improvements below.

You may also notice other changes to the manuscript, some of which were done in response to comments from the editor and other reviewers, and others were implemented to meet strict formatting requirements of the journal (including changes to the title and abstract, and the removal of footnotes and italics). We think the end result is a much-improved manuscript, though at times we struggled to balance the desire to thoroughly address reviewers' comments with the space constraints of the journal.

Thanks again for taking the time to review this manuscript!

Comment 1:

Overall I think this is an excellent manuscript, it makes an important contribution and is competently executed. Before recommending publication however, I would like to see the authors address three issues (in falling order of importance).

First, I would like for the authors to address the loss of 47% of the violent events in the UCDP data. The authors do an admirable job of discussing and convincingly addressing a range of other inferential problems, but the potential threats to valid inference posed by this non-random loss of data are not addressed. It should be, because it is at least possible that it affects the findings. The reason they discard 47% of the violent events is because they are insufficiently precise with regards either to location or day. Here is how this might affect the findings and how we understand them:

Precision is going to be related to the core – periphery dimension, including presence/absence of cell towers and capital vs noncapital districts (Cf Weidmann 2014).

The threshold for reporting (esp with sufficient detail to be coded with enough precision to be included in the present study) is lower in core areas, and higher in peripheral areas (Cf Galtung & Ruge 1965, Öberg & Sollenberg 2011).

So what are some of the things that might lift an event in a peripheral area over the threshold and increase the likelihood that it is reported (with sufficient precision)? Events that represents a new development in a conflict, a break with a previous pattern, such as when control of an area changes hands, or when a new (and esp news worthy) actor makes advances/establish themselves or become active in new areas (Cf Öberg & Sollenberg 2011).

Thus, of the reported events with sufficient precision to be included you are likely to have a higher proportion of “business as usual”/typical events in capital districts (and other more core areas) and a higher proportion of events associated with new and more threatening developments in peripheral areas. The bar for something being reported (esp with geographic precision) is higher in peripheral areas, so there will in peripheral areas be a selection towards more newsworthy events like the appearance of new actors, the territorial advance/retreat of some actor, or the loss of control by the government (or the seeming threat thereof) – all of which may imply higher levels of threat than otherwise equivalent violence and higher levels of threat may generate more displacement.

This might explain or be part of the explanation for the findings in Figure 3d provincial capitals vs non-capitals. It could possibly effect other findings also, like those on IS and on previous violent events, but that is less obvious.

While there is little that can be done about the lack of geographic precision, I think the authors need to take serious potential problems it raises and at least discuss potential implications.

To understand the problem I suggests the authors consult the sources below (they also give references to potential further readings).

On how reporting is related to core-periphery using military SIGACTS data for Afghanistan and comparing it to an earlier version of the UCDP data:

Nils B. Weidmann 2014. "On the accuracy of Media Based Conflict Event Data" *Journal of Conflict Resolution* 59(6): 1129-1149. <https://doi.org/10.1177/0022002714530431>

On how news evaluation works and might affect what gets reported and from where (i.e. what is newsworthy in different contexts):

Galtung, Johan, Ruge, Mari Holmboe. 1965. "The Structure of Foreign News: The Presentation of the Congo, Cuba and Cyprus Crises in Four Norwegian Newspapers." *Journal of Peace Research* 2 (1): 64–91.

Öberg, Magnus, Sollenberg, Margareta. 2011. "Gathering Conflict Information Using News Resources." In *Understanding Peace Research: Methods and Challenges*, edited by Höglund, Kristine, Öberg, Magnus, 47–73. New York: Routledge.

Thank you for this excellent and insightful observation, and for providing very helpful references. We agree that this is an important issue that was not sufficiently discussed in the prior manuscript. The revised paper makes several changes to acknowledge and address this limitation.

First, we now flag very early in the paper (i.e., in the *Introduction*) that there are issues to consider with missing violence data. The relevant text reads, "We use approximately six thousand events in our analysis, corresponding to the subset of all events that are recorded with sufficient spatial and temporal precision (53% of all recorded events); the limitations of this data source are discussed in the *Limitations* section of this paper.

Second, we have created a new “Limitations” section in the paper, and have added discussion of the key points you make above. We thank you again for highlighting the specific core/periphery issue, and for the references -- both of which we have incorporated into this discussion. The relevant text reads, “We also note limitations of the data we use to measure violence and conflict. These data are collected from media sources, supplemented by NGO and IGO reports, field reports, and books (Sundberg and Melander 2013). But not all events are included in media reports; inclusion is affected by a large number of factors (Öberg and Sollenberg 2011). For example, more populous regions and economic centers have a larger media presence and are thus better covered than peripheral areas; contested areas and places with less infrastructure tend to have poorer coverage. Media coverage is also influenced by how “newsworthy” the event is likely to be (Galtung and Ruge 1965). Our analysis is further restricted to those events for which the date and location (i.e., the district) are known. Since the spatial and temporal precision of event reporting is not random (Weidmann 2015), and that non-randomness may even differ across regions of Afghanistan, this could introduce bias into our analysis. For instance, if the threshold for reporting violence in rural areas is higher than the threshold in urban areas (Galtung and Ruge 1965, Öberg and Sollenberg 2011), that could bias our analysis to finding larger effects of violence in rural areas. We perform robustness tests in *Materials and Methods* (Section J) to assess the likely magnitude of this bias, but we also acknowledge that this is a limitation of our data -- and a limitation shared by most empirical work on conflict.”

Finally, to assess the potential biases introduced by our removal of violent events that are not recorded with sufficient (district-level) precision, we performed additional analysis to determine whether the spatial precision with which an event is recorded is correlated with the magnitude of the displacement effect. Specifically, we estimate separate coefficients for each of the three types of events that we use in our analysis, based on their available geographic precision: (A) Events in which the exact location of the event is known and coded (N=1698), (B) Events which occurred within a 25 km radius around a known point (N=789) and (C) Events in which only the district is known (N=969). (This analysis is imperfect, because we would ideally want to know what the response is for events with location not known to a district-level precision (i.e., the events not included in our paper), but of course we cannot do that analysis with the data we have.)

This new analysis is presented in Section J (Implications of missing violence data) of *Materials and Methods*, and Figure S6 presents the results, using an analogous empirical strategy as that used to construct Figures 2a and 3a-d in the manuscript. If it is indeed the case that events recorded with lower precision represent less threatening events, then we might expect coefficients for (C) to have smaller magnitudes than that for (A) and (B). This does not appear to be the case: there are differences in the point estimates, but the confidence intervals of the estimated coefficients for the three types of events overlap.

We acknowledge that this analysis does not eliminate the concerns you raise above, and it doesn't directly address the specific core/periphery concern. However, it is reassuring that the magnitudes and overall conclusions that we draw in our main analysis are not substantively changed by analyzing these separate groups of events.

The relevant text in Materials and Methods (Section J) reads:

As discussed in Limitations, our analysis does not include violent events that are not associated with specific locations (i.e., where we do not know the district in which the event occurred). This could introduce bias into our analysis if certain types of violence (with specific migration responses) are systematically more or less likely to have known locations. We therefore conduct additional analysis to determine whether the spatial precision with which an event is recorded is correlated with the magnitude of the displacement effect.

Specifically, using the same empirical approach described in Section G, we create separate treatment indicators for each of the three types of events that we use in our analysis, based on their available geographic precision: (A) Events in which the exact location of the event is known and coded, $A_{\{d,t+\tau\}}$ (N=1698), (B) Events which occurred within a 25 km radius around a known point, $B_{\{d,t+\tau\}}$ (N=789), and (C) Events in which only the district is known, $C_{\{d,t+\tau\}}$

$$\begin{aligned}
 &g(\mathbb{E}(Y_{dt,k}|X_{dt}, A_{d,t+\tau}, B_{d,t+\tau}, C_{d,t+\tau})) \\
 &= \gamma_d + \lambda_t + \sum_{\tau=-30}^{180} \beta_{A,\tau} A_{d,t+\tau} + \sum_{\tau=-30}^{180} \beta_{B,\tau} B_{d,t+\tau} + \sum_{\tau=-30}^{180} \beta_{C,\tau} C_{d,t+\tau} \quad (5)
 \end{aligned}$$

(N=969). These replace the treatment indicators in (1):

The results, shown in *SI Appendix Figure S7*, indicate that the displacement response is very similar for violent events with these three different levels of spatial precision. There are small differences in the point estimates but the general pattern of the response is unchanged, and the confidence intervals of all three violence types overlap. Of course, this analysis does not eliminate the possibility

that there might be a qualitatively different displacement response to violent events that are not recorded in our dataset (or for which district information is unknown). Unfortunately, we cannot directly test that concern, since we cannot estimate the displacement effect of violence when the location of the violence is not known.

Comment 2:

Second, scope conditions. How far do the findings in this manuscript travel? Are the findings particular to, or contingent on some aspect of this context, or can we expect similar patterns to obtain elsewhere? As a conflict country Afghanistan may in some ways be an outlier.

Afghanistan have a 40+ year history of war and displacement. Millions of people have left, and it seems likely that 40 years of fighting has led to some social adaptations and strategies for coping with violence and threats of violence that are not present in other contexts. They have several ethnic and sectarian divisions layered on top of a partly disrupted clan based society all of which is reflected in the armed groups. And there is much foreign involvement and interference. So, for example, does the tendency to flee to district capitals have something to do with clan and ethnic or sectarian divisions (where one can safely flee), or is it because of safety in numbers, or is it because of stronger

government control/presence in the district capitals as the authors suggest? Presumably the idp's seek some measure of safety, but government control does not always imply safety, even if it means less fighting. Does the age structure affect the propensity to flee? The median age in Afghanistan is just over 18 years and it is one of the poorest countries on the planet. Would patterns be similar in a country with a different age structure and more resources? These are not the only questions and I do not know if they are necessarily the most salient ones so I do not expect answers to them specifically. But I think some mention or discussion of what the authors think about the scope conditions of their findings are in order. It is valuable for future research. It is also, I think, important for ethical reasons to at least flag clearly if the authors are not able to delineate the scope conditions so that policy makers and others who may act of these findings in other settings are

aware that they may not apply - at least not equally - in other cases.

Thank you for these questions and suggestions. We agree that this is an important and useful discussion, and have included a paragraph in *Limitations*. Briefly, we believe the general result that displacement is affected by violence is likely to be pervasive, but specific results (including magnitudes and elements of heterogeneity) may not generalize, due to some of the factors that you rightfully suggest. Some specific examples that we cite include Afghanistan's long history of conflict resulting in different adaptations, and location-specific results that are potentially influenced by many context-specific factors. We have taken the reviewer's suggestion to urge policy- and other decision- makers to be cautious when interpreting and extrapolating from our findings. The relevant text reads, "However, the Afghan setting is unique in many respects, and some results (such as the

magnitude of the estimated effect, or specific elements of the heterogeneous response) may be specific to Afghanistan. For example, Afghanistan has a decades-long history of conflict; our results will likely translate better to settings with endemic conflict than ones where violence is new. Afghanistan also has strong clan-based, ethnic, and sectarian divisions; a strong patriarchal society; and a relatively poor and young population. These

contextual factors likely impact migration decisions in complex ways; our results will naturally be most relevant to environments with similar characteristics.”

Comment 3:

Finally, the subject of this study is the displacement of Afghan people in Afghanistan in the internationalized but still very much Afghan civil war. The war in Afghanistan pre-dates the current US intervention by a quarter of a century or so. Yet, in the abstract a justification for looking at Afghanistan seems to be that it is the longest war in US history. While this may be true, it strikes me as irrelevant to the study and bizarrely US-centric. I would suggest the authors simply drop that part.

Thank you for pointing this out. We agree and have replaced this text. The sentence now reads, “We develop an approach to measure the impact of violence on internal displacement using anonymized high-frequency mobile phone data. We use this approach to quantify the short- and long-term impact of violence on internal displacement in Afghanistan, a country that has experienced decades of conflict.”

R3’s comments are in black/Arial; our point-by-point responses are in blue/Times

Thank you very much for your careful and expert review of our manuscript, and for the detailed and constructive suggestions. You will find point-by-point response to each of your Suggested Improvements below.

You may also notice other changes to the manuscript, some of which were done in response to comments from the editor and other reviewers, and others were implemented to meet strict formatting requirements of the journal (including changes to the title and abstract, and the removal of footnotes and italics). We think the end result is a much-improved manuscript, though at times we struggled to balance the desire to thoroughly address reviewers’ comments with the space constraints of the journal.

Thanks again for taking the time to review this manuscript!

Comment 1:

In this paper, the authors use mobile phone traces to measure internal displacement within Afghanistan in response to violent events. The authors are able to quantify where people are displaced to and for how long they remain displaced with respect to their home location. All locations are defined to district (admin 2) level. The authors report that there is a causal and significant increase in out-displacement following (and to a lesser extent preceding) a violent

event. The nature of the displacement is affected by the group responsible for the attack, the severity and frequency of the violence as well as the proximity to urban/rural areas.

I believe that this paper is a notable and high quality contribution to the scientific literature on conflict, migration and the use of non-traditional data sources for official statistics. I would also add that this work is not a pure academic work but closer to 'scholar-practice' and so it should be reviewed with that in mind.

What this means concretely is the following; a large part of the contribution here is not concerned with methodological advancement but with the enormous work that goes into a complex partnership with a mobile phone operator in a low income country. To navigate the regulations of accessing this data in a responsible way, dealing with data cleaning and bringing to publication the study of such a sensitive topic is quite an achievement. Part of the value is the precedent that this sets for further similar studies and opportunities for other researchers and partnerships and hopefully, eventually, the operationalisation of this kind of work into sustainable partnerships between governmental organisations and mobile phone operators.

The flipside of this is that this study will surely be read by many technically literate and skeptical non-academics who work in National Statistical Agencies and similar. As a result, speaking from deep experience, any far reaching claims made by the authors could potentially be damaging to this endeavour in the long term.

With this in mind, my main substantive request to the authors is to be more clear about the limitations of this work. This is not a criticism of the work itself, it is simply the nature of the events and the data.

We thank the reviewer for the kind remarks. We have reorganized the *Discussion* section to include a specific *Limitations* section, and expanded our discussion significantly based on feedback from the reviewers.

Comment 2:

-- Figure 1 is the most critical of the entire paper: the spatial distribution of cell towers and violent events. Anyone who works with mobile phone data knows that cell towers are heterogeneously distributed and it will surprise no-one that there are parts of Afghanistan with no cell phone coverage. So please break out this figure into 3 parallel maps showing (i) cell tower locations (ii) violent events and (iii) district population so that these can be easily compared.

Thank you for this suggestion. We have included the requested figure as Supplementary Figure 1. In the main text, we reference this figure together with Figure 1.

Comment 3:

-- It is common practice to evaluate the user base in mobile phone studies by a simple comparison of district population from official statistics and mobile subscriber home location. Of course the best official statistics in Afghanistan will be imperfect but this kind of validation is important and will increase the impact of the work in the long term.

Thank you for this suggestion. We performed the suggested analysis by comparing district population as estimated by Afghanistan's Central Statistical Office from 2015-2016 to the mean number of subscribers having home locations assigned to each district, averaged over the same one-year period. The resulting scatterplot is shown in the figure below; the Pearson's correlation coefficient is .94. Excluding Kabul (the outlier on the top-right), the correlation coefficient is .68.

This is the corresponding figure on a log scale; the correlation coefficient is .53.

We now reference this new analysis in *Materials and Methods* Section C: “In Afghanistan, we find that the geographic distribution of daily modal locations of mobile phone subscribers broadly reflects the geographic distribution of the population. In particular, when comparing the number of mobile phone users in each district to the district population as estimated by Afghanistan's Central Statistical Office, the Pearson's correlation coefficient is 0.94 (95% CI: (0.92, 0.95), $P < .001$). (We calculate the number of subscribers in each district as the number of subscribers whose daily modal location is assigned to that district, averaged over all days in which the district has a non-zero number of subscribers, for a one-year period. Official estimates are obtained from <https://data.humdata.org/dataset/estimated-population-of-afghanistan-2015-2016>. On a log scale, the Pearson's correlation is 0.53 (95% CI: (0.43, 0.61), $P < .001$).”) To keep the manuscript short, we have not currently included the figure in the manuscript, but we are happy to do so if the reviewer thinks this is important.

Comment 4:

-- Again, common practice in mobile phone studies is to construct a Voronoi tessellation of cell towers and to use the overlap of the Voronoi cells with the administrative districts to, perhaps probabilistically, assign a home location. It's not clear to me from the provided details and citation (A general approach to detecting migration events in digital trace data) what the authors did instead and why. Some more explanation here would be welcomed.

We apologize for the lack of clarity in how we assign home locations. We have revised and expanded the relevant discussion in *Materials and Methods* Section C (“Measuring migration”) to provide more detail. This section now reads:

Our first step is to derive a ‘daily modal location’ for each subscriber for each day, which is intended to capture the district in which the subscriber spends the majority of their time on that day. For each individual, we first compute their most commonly used cell tower in each hour. Then, for each 24-hour period from 6 A.M. to 6 A.M. the next day, we compute the mode of the hourly modal towers. The towers are then mapped to districts using point-in-polygon assignment. Similar methods have been used and validated in other work (Phithakkitnukoon et al. 2011, Toomet et al. 2015, Hong et al. 2019, Hankaew et al. 2019, Deville et al. 2014, Vanhoof et al. 2018). While several prior studies use night-time hours to infer daily locations; we instead use all hours, which allows us to include more individuals in our analysis. For example, in April 2013, data are available for approximately 31 million individual-days using night-time hours (6 P.M. to 6 A.M.), while 61 million individual-days are defined when using all hours (6 A.M. to 6 A.M.). The two approaches are highly correlated: Of the 31 million observations available using night-time hours, 89% record the same daily modal districts when computed using all hours. Another common approach in the literature divides the physical terrain into approximate catchment areas of each cell tower, using a Voronoi tessellation (e.g., Gonzalez et al. 2008, Deville et al. 2014). Our analysis focuses on slightly larger administrative districts, since many of our violent events are only identified at the district level.

In Afghanistan, we find that the geographic distribution of daily modal locations of mobile phone subscribers broadly reflects the geographic distribution of the population. In particular, when comparing the number of mobile phone users in each district to the district population as estimated by Afghanistan's Central Statistical Office, the Pearson's correlation coefficient is 0.94 (95% CI: (0.92, 0.95), $P < .001$). (We calculate the number of subscribers in each district as the number of subscribers whose daily modal location is assigned to that district, averaged over all days in which the district has a non-zero number of subscribers, for a one-year period. Official estimates are obtained from <https://data.humdata.org/dataset/estimated-population-of-afghanistan-2015-2016>. On a log scale, the Pearson's correlation is 0.53 (95% CI: (0.43, 0.61), $P < .001$.)

These daily modal locations tend to be sparse and noisy: for instance, many people do not use their phones on every single day; people may take short trips to nearby (non-residential) locations, and so forth. Thus, our second step employs an unsupervised scanning algorithm (Chi et al. 2020) to identify contiguous segments in which a subscriber is, with high probability, resident in a single district. This algorithm helps smooth the influence of noise (for instance, long periods where a person is primarily in one location, but intermittently visits other

locations for one or two days) and missing data (for instance, when a person uses their phone infrequently, but almost exclusively from the same location), and has the advantage of not arbitrarily grouping days into calendar weeks or calendar months.

You also raise the valid point that an alternative approach to inferring locations would probabilistically assign people, based on their cell tower, to administrative districts. Note that our current point-in-polygon approach is almost identical to deterministically assigning people, based on their cell tower, to the administrative district with which that tower has the most overlap (it will not be identical if districts are not convex, but in practice this is rare). We do not take the probabilistic approach because it would significantly complicate our panel event study estimation framework, which requires that each individual be associated with a single district on each day. There are likely alternative modeling approaches that would not have this limitation, but we are reluctant to change our workhorse model at this stage in our project.

Comment 5:

-- An overarching issue in studies of conflict and other extreme events is that they are very sparse. The case of Afghanistan is no different, by my estimate there are roughly $365 \times 4 \times 398 \sim 600k$ district-days in the data, with $\sim 3k$ violent events with district information giving a frequency of $\sim 0.5\%$. This may have some consequences for the validity of the causal analysis, I'm not qualified to comment on that, however at the very least the authors should make some reference to this aspect of the data and what confounding effects it might introduce.

Thank you for raising this concern. In the manuscript, we now explicitly note the sparsity of the violence data. We also clarify that the statistical model we employ is robust to sparsity; as long as events are recorded accurately, estimates will not be biased because of this. The relevant text reads, "although the treatments (violent events) occur relatively infrequently, the statistical model we employ is robust to sparsity; if all of the events are recorded accurately, estimates will not be biased because of sparsity."

However, if the sparsity is not random, and if specific types of events are systematically missing, that could bias our estimates -- this is the focus of your next comment, so we will respond to that concern below. We also state this in the manuscript, "Non-random missingness of recorded events could bias estimates and are discussed in *Limitations*."

Comment 6:

-- Relatedly, the authors report that only 47% of the ground truth data on violent events have district level spatial accuracy. Again, this is no fault of the authors but should be addressed

clearly. In fact, the use of passively collected data for proxying measures of interest is increasingly held back by a lack of high quality data against which to validate. The authors should comment on (i) what the elimination of roughly half the violent events might mean for their results and (ii) how these shortcomings in conflict data might be addressed in the future. I can recommend the following paper as an entry point into this debate; Growing pains for global monitoring of societal events (2016), Wang et al (<https://science.sciencemag.org/content/353/6307/1502.summary>)

Thank you for pointing this out and for the suggested reference. We agree that this is an important issue. The revised manuscript is now more explicit about this limitation. For instance, the introduction now notes, “We use approximately six thousand events in our analysis, corresponding to the subset of all events that are recorded with sufficient spatial and temporal precision (53% of all recorded events); the limitations of this data source are discussed in the *Limitations* section of this paper.”

We have also added a new section on “Limitations”, which includes a discussion of this issue: “We also note limitations of the data we use to measure violence and conflict. These data are collected from media sources, supplemented by NGO and IGO reports, field reports, and books (Sundberg and Melander 2013). But not all events are included in media reports; inclusion is affected by a large number of factors (Öberg and Sollenberg 2011). For example, more populous regions and economic centers have a larger media presence and are thus better covered than peripheral areas; contested areas and places with less infrastructure tend to have poorer coverage. Media coverage is also influenced by how “newsworthy” the event is likely to be (Galtung and Ruge 1965). Our analysis is further restricted to those events for which the date and location (i.e., the district) are known. Since the spatial and temporal precision of event reporting is not random (Weidmann 2015), and that non-randomness may even differ across regions of Afghanistan, this could introduce bias into our analysis. For instance, if the threshold for reporting violence in rural areas is higher than the threshold in urban areas (Galtung and Ruge 1965, Öberg and Sollenberg 2011), that could bias our analysis to finding larger effects of violence in rural areas. We perform robustness tests in *Materials and Methods* (Section J) to assess the likely magnitude of this bias, but we also acknowledge that this is a limitation of our data -- and a limitation shared by most empirical work on conflict.”

The above text references a new section, “Implications of missing violence data”, that provides additional empirical analysis. The main result of that analysis is that we do not see a correlation between the magnitude of the migration response and the precision of spatial information for the violent event. We take this as encouraging -- but not conclusive -- evidence that heterogeneous spatial accuracy is not heavily influencing our results.

Finally, we have added a brief discussion of what we hope for the future for conflict events data. The relevant text reads, “While these problems of exclusion are faced by all

violent events data sets that rely on news sources, and there is little we can do to improve the quality of the data, we are optimistic that the quality of conflict event data will improve as they gain more widespread use. For instance, several papers now articulate best practices for the collection of conflict data (Salehyan 2015, Wang et al. 2016), and researchers are increasingly exploring how social media and other non-traditional data can be used to source conflict data (Zeitsoff 2011).”

Comment 7:

Finally, I would be interested in the authors perspective on the degree to which this is a descriptive analysis vs a predictive endeavour. I would imagine that there is a delay in reporting and compiling violent events in the Uppsala database and similar. Would it be possible to use mobile phone derived measures of displacement as an early warning of a violent event (particularly in light of the previous points regarding sparsity and false positives)?

This is an interesting point that we considered early on in this project. In principle, we agree that this might be feasible, and we are aware of one recent paper that explores this in detail (Berger et al. 2014). However, as you might imagine, this is a complex and subtle problem, and to properly address it would require serious investigation. For this reason, we decided it is out of scope for the current paper, although we now mention this in the discussion as an interesting dimension: “Our analysis also indicates that people appear to anticipate the occurrence of violence , leaving before it occurs --- a finding that relates to prior work on the predictability of violence and conflict (Ward et al. 2010, Hegre et al. 2013, Cederman and Weidmann 2017, Bazzi et al. 2021, Kapoor and Narayanan 2021), including recent work using mobile phone data (Berger et al. 2014).”

Decision Letter, second revision:

3rd November 2021

Dear Josh,

Thank you once again for your manuscript, entitled "Mobile Phone Data Reveal the Effects of Violence on Internal Displacement," and for your patience during the peer review process.

Your manuscript has now been evaluated by the same 3 reviewers who saw the original version of your study. We also recruited a new reviewer (Reviewer 4) with additional expertise in civil conflict, particularly in Afghanistan. You will find all reviewer comments included at the end of this letter. Although 3 reviewers are satisfied with your revision and the changes made, Reviewer 4 raises some important remaining concerns. We remain interested in the possibility of publishing your study in *Nature Human Behaviour*, but would like to consider your response to these concerns in the form of a revised manuscript before we make a decision on publication.

In particular, we ask that you revise to address Reviewer 4's concerns. Please ensure that your revision thoroughly considers and discusses the mechanisms this reviewer highlights, including those related to selection effects, anticipation effects, and the move towards provincial capitals. We also request that you carry out the additional robustness check using ACLED recommended by the reviewer.

In addition to these core concerns, Reviewer 4 also highlights the question of data availability. Our journal policy on data availability recognizes that there are circumstances in which data cannot (and sometimes should not) be made publicly available. However, we do require that data in principle be available to future researchers, subject to necessary conditions. We understand from your data

availability statement that there is in principle a process through which other researchers could obtain the data from the provider. If at all possible, we would encourage you to go a step further by revising your data availability statement to allow for access of the full dataset to accredited academic researchers upon reasonable request and with permission from the provider. We encourage you to consult with the data provider to see whether this type of access arrangement would be permissible. Of course, we entirely understand that the data provider may not agree and may want to remain the sole provider of the data. In this case, we would ask you to confirm with the provider that they would share the same dataset with accredited academic researchers upon reasonable request. (Note that we do not require naming the provider in the data availability statement.) If you have any questions about our data availability policies, please don't hesitate to let me know.

Finally, your revised manuscript must continue to comply fully with our editorial policies and formatting requirements. Failure to do so will result in your manuscript being returned to you, which will delay its consideration. To assist you in this process, I have attached a copy of our checklist that lists all of our requirements. If you have any questions about any of our policies or formatting, please don't hesitate to contact me.

In sum, we invite you to revise your manuscript taking into account all reviewer and editor comments. We are committed to providing a fair and constructive peer-review process. Do not hesitate to contact us if there are specific requests from the reviewers that you believe are technically impossible or unlikely to yield a meaningful outcome.

We hope to receive your revised manuscript within four to eight weeks. We understand that the COVID-19 pandemic is causing significant disruption for many of our authors and reviewers. If you cannot send your revised manuscript within this time, please let us know - we will be happy to extend the submission date to enable you to complete your work on the revision.

- Include a "Response to the editors and reviewers" document detailing, point-by-point, how you addressed each editor and referee comment. If no action was taken to address a point, you must provide a compelling argument. This response will be used by the editors to evaluate your revision and sent back to the reviewers along with the revised manuscript.
- Highlight all changes made to your manuscript or provide us with a version that tracks changes.

[REDACTED]

We look forward to seeing the revised manuscript and thank you for the opportunity to review your work. Please do not hesitate to contact me if you have any questions or would like to discuss these revisions further.

Sincerely,
Aisha

Aisha Bradshaw, PhD
Senior Editor
Nature Human Behaviour

Reviewer expertise:

Reviewer #1: political science, civil conflict

Reviewer #2: political science, civil conflict

Reviewer #3: network science, mobile phone data analysis

Reviewer #4: political science, conflict in Afghanistan

REVIEWER COMMENTS:

Reviewer #1:

Remarks to the Author:

The authors should be commended for an extremely thorough response to reviewers and for their superb revisions. I have learned a great deal from this paper. I highly recommend publication.

Reviewer #2:

Remarks to the Author:

I am happy with the way the authors have addressed the concerns expressed in my previous review and now recommend publication.

Reviewer #3:

Remarks to the Author:

I thank the authors for their comprehensive and detailed rebuttal to mine (and other reviewers') comments. I am happy to support publication of the revised manuscript.

Reviewer #4:

Remarks to the Author:

Review of "Mobile Phone Data Reveal the Effects of Violence on Internal Displacement"

This short manuscript uses high-frequency phone logs to digitally trace the effects of violence on the migration of Afghans across districts in Afghanistan for a four year period. The authors find that violence (especially that by IS-K) is associated with an increased likelihood that individuals (or, at least, those with phones) will migrate from the district toward provincial capitals. The authors also find that individuals appear to anticipate violence and will migrate several days before the recorded violent incident (see below).

I've had the chance to read the revised manuscript and the extensive revision memo. I was not an original reviewer. I believe that the paper makes a technical, if narrow, methodological contribution that will be valuable for other individuals (and agencies) seeking to track the movement of these populations in wartime. While the central finding is perhaps unsurprising --- few would contest the violence toward civilians is an inducement to their migration from an area --- the manuscript offers a useful blueprint for

others seeking to use digital tracing in wartime and other environments (see Christia et al, 2021, on similar use of cell phone records to track migration from drone strikes in Yemen).

I believe the authors were responsive to the questions/comments raised by the reviewers. I had only a few relatively minor concerns and questions.

1. What is the justification for using 180-day and 30-day windows to estimate the effects of violence? Are the results sensitive to variation in the length of these lagging and leading terms?

2. As the authors note, reporting on violence isn't random. But neither is the application of violence. There's little discussion here of how selection effects (that is, the targeting of a particular district) might drive some of these findings. Are the Taliban (and in particular, IS-K) targeting most likely villages/areas for displacement, and so we're actually seeing the upper bound on migration? Or should we consider the violence as-if random, and so we're getting a credible estimate of the effects of violence on migration? The magnitude of the effect isn't discussed much here, and it's hard to get a sense of whether this is a large effect or a modest one. This is partly an empirical question but also a theoretical one: if the Taliban are purposefully trying to induce migration in an area (to chase out government supporters, for example), then this would suggest that individuals are fairly resilient in the face of violence. But if this violence is haphazard, then it suggests a greater underlying degree of sensitivity (since those most likely to move are not being singled out). How should we interpret the magnitude of the effect and the severity of selection bias?

3. The finding that individuals appear to anticipate violence is largely due to the fact that Afghan forces and the Taliban routinely warned civilians of major military operations (the kind most likely to be captured by the GED dataset). Civilians were often given a specified time to leave their villages before the operation began. As a result, the pre-violence migration observed in the findings is likely due less to civilian networks and other (unrecorded) violence than to efforts by the ANDSF and Taliban to warn villagers via leaflets, word of mouth, and SMS messages.

4. I'd note an additional explanation for why villagers headed to the provincial capitals: that's where Western and Afghan NGOs had the best distribution networks for humanitarian aid. In line with some of these findings, it's apparent that the effectiveness of aid after civilian casualties was sensitive to location (including proximity to bases and to larger urban centers; see Lyall 2019), so it makes sense that individuals head to these locations.

5. The GED has a high degree of missingness for Afghanistan. Have the authors tried using ACLED as a robustness check? It typically has a higher degree of coverage for Afghanistan than Uppsala.

6. One final note. I applaud the authors for being able to get these data (presumably from **[REDACTED]**, though the telecom is never identified). I am concerned, however, that these data will not be made available for replication purposes to other scholars. A "small subset of the simulated data" is not especially helpful for replication. I'm not sure how to square this circle, but withholding the data seems at odds with our renewed emphasis on data transparency and availability.

Author Rebuttal, second revision:

R4's comments are in black/Arial; our point-by-point responses are in blue/Times

Thank you for your extremely thorough and constructive review, which has helped us to further improve our manuscript. We have provided a point-by-point response to each of your Suggested Improvements below.

Comment 1:

What is the justification for using 180-day and 30-day windows to estimate the effects of violence? Are the results sensitive to variation in the length of these lagging and leading terms?

Thank you for this question; we agree that this should be discussed in more detail. We focus on this particular window (-30 to +180) because it captures the primary period during which significant effects are observed.

The results do not change when we use a longer window. This can be seen below. First, we reproduce the main figure (2b) from the paper, which shows 30-day displacement from 30 days before to 180 days after violent events. We then show the same figure (not in the paper) that includes the period from 60 days before to 365 days after violent events. In the longer window, the effects with larger lags are noisier (because there is much less data available to identify those effects) and largely statistically insignificant.

Reproduction of Figure 2b (-30 to +180):

Results using a longer window (-60 to +365):

When we use a shorter period, the results are very similar to those shown in the main text. For instance, below we show the results when estimated using a period from 30 days before to 30 days after violence.

Results using a shorter window (-30 to +30):

We have added this explanation in *Materials and Methods* Section E: “Lags of $\tau \in [-30, 180]$ are used, representing violence in the district 30 days in the future to 180 days in the past. This window was chosen as all effects were observed to lie within this window; results are insensitive to a longer window length, while shorter windows are unable to capture all effects of interest.”

Comment 2:

As the authors note, reporting on violence isn't random. But neither is the application of violence. There's little discussion here of how selection effects (that is, the targeting of a particular district) might drive some of these findings. Are the Taliban (and in particular, IS-K) targeting most likely villages/areas for displacement, and so we're actually seeing the upper bound on migration? Or should we consider the violence as-if random, and so we're getting a credible estimate of the effects of violence on migration? The magnitude of the effect isn't discussed much here, and it's hard to get a sense of whether this is a large effect or a modest one. This is partly an empirical question but also a theoretical one: if the Taliban are purposefully trying to induce migration in an area (to chase out government supporters, for example), then this would suggest that individuals are fairly resilient in the face of violence. But if this violence is haphazard, then it suggests a greater underlying degree of sensitivity (since those most likely to move are not being singled out). How should we interpret the magnitude of the effect and the severity of selection bias?

Thank you for this question and comment. Indeed, a key identifying assumption of our model is that there are no unobserved time-varying confounds, meaning that we assume the precise day in which a district experiences violence is as-if random, conditional on district and time fixed effects, and the occurrence of violence in the surrounding window of time (Equation 1).

Qualitatively, we find this assumption plausible because the precise timing (i.e., the day on which violence occurs) of insurgent attacks is often meant to surprise government forces. However, since violent events are not experimentally assigned, the occurrence is not truly random, and selection effects such as those that you describe may affect our results (and how we interpret them).

In the revision, we have included several new tests to assess the randomness of violence. In particular, we focus on understanding whether the occurrence of violence can be predicted beyond what is captured in our econometric model (Equation 1). In other words, we seek to understand whether violence can be predicted beyond what an econometrician would predict given information about district and time fixed effects, and violence in the surrounding period.

Specifically, we first regress the occurrence of violence on day t on the control variables in Equation 1:

$$g(E(T_{dt} | \gamma_d, \lambda_t, T_{d,t+\tau})) = \gamma_d + \lambda_t + \sum_{\tau \in [-30, 180] \setminus \{0\}} \beta_\tau T_{d,t+\tau}$$

From this, we obtain the residuals $T_{dt} - \hat{E}(T_{dt})$, which are a measure of unexpected violence.

We then assess whether these residuals can be predicted using other potentially relevant time-varying environmental information. Specifically, we seek to predict the residuals using recent lags and trends in the outcome variable (30-day displacement), as well as information about the number of individuals in a district. We focus on four different sets of predictor variables:

- (1) 1-15 day lags of the 30-day displacement outcome $Y_{dt,30}$
- (2) 1-15 day lags of the number of subscribers observed to be in district d at time $t-30$, i.e., 31-45 days ago
- (3) Trends of (1), i.e., $\Delta x_{t-1} = x_{t-1} - x_{t-2}$, $\Delta x_{t-2} = x_{t-2} - x_{t-3}$, etc.
- (4) Trends of (2).

As a benchmark specification, we test whether residual violence can be predicted with a linear model of those predictor variables. Separately, we also test a more flexible machine learning approach (i.e., a random forest with 50 trees and 10-fold cross validation) to predict residual violence. Across all tests, we find that these characteristics are consistently unable to predict residual violence ($R^2 \leq 0.000277$). These results, included as *Supplementary Table S1* in the revision (reproduced below), suggest that violence is not predictable beyond what is accounted for in our preferred econometric specification.

Supplementary Table 1: Can violence be predicted?

	(1)	(2)	(3)	(4)	(5)
Panel A: Model performance					
Linear model R^2	0.000031	0.000029	0.00024	0.00022	0.00028
Random forest R^2	0.00000014	0.0000099	0.000081	0.000020	0.000019
Panel B: Variables used to predict residual violence					
Lags of displacement (lags 1-15)	X				X
Trends of displacement		X			X
Lags of #subscribers (31-45 days ago)			X		X
Trends of #subscribers				X	X

Notes: Table indicates R^2 of 10 different models that attempt to predict residual violence, where residual violence is defined as the residuals obtained by regressing violence on the control variables in our main Equation (1). Specifically, we first regress the occurrence of violence on day t on the control variables in (1), i.e., $g(\mathbb{E}(T_{dt}|\gamma_d, \lambda_t, T_{d,t+\tau})) = \gamma_d + \lambda_t + \sum_{\tau \in [-30, 180] \setminus \{0\}} \beta_\tau T_{d,t+\tau}$, and obtain the residuals $T_{dt} - \hat{\mathbb{E}}(T_{dt})$. We then attempt to predict those models using the variables listed in Panel B, where ‘lags of displacement’ refers to 1-15 day lags of the 30-day displacement outcome $Y_{dt,30}$, and ‘lags of number of subscribers’ refers to 1-15 day lags of the number of subscribers observed to be in district d at time $t - 30$, i.e., 31-45 days ago. Trends refer to using differences of the displacement outcome and number of subscribers as predictors: i.e., $\Delta x_{t-1} = x_{t-1} - x_{t-2}$, $\Delta x_{t-2} = x_{t-2} - x_{t-3}$, etc. Each column of Panel A reports the performance (R^2) of two different models: a linear regression (first row) and a 10-fold cross-validated random forest (second row).

Additionally, if the Taliban or IS-K are specifically targeting an area to displace people, this might be captured in a time-varying fixed effect at a coarser scale; the robustness check in *SI Appendix Figure S5* shows that after adding these more restrictive region*month fixed effects to Equation 1, the main results are qualitatively unchanged.

Taken together, we do not find evidence against the assumption of as-if random violence, suggesting that the magnitude of the effects of violence on displacement that we estimate are credible. You also raise a point about the *intent* of the violence, i.e., whether it is specifically to displace or otherwise. We see this as a slightly different issue from the randomness or predictability of the violence: our earlier analysis suggests that violence is not predictable, however we do not distinguish violent events by their motive, so when we say that the effect of violence is to increase the odds of displacement by 3% (say), we can think of this as the average over both violence targeted at displacing people, and other types of violence. However, in light of the above analysis, we consider both of these types of violence to be “haphazard,” to the extent that we do not find any evidence that violence is predictable.

To make these points clear, the revision includes a section in *Materials and Methods* Section E that describes this new analysis:

The key identifying assumption that there are no unobserved time-varying confounders requires that the precise day in which a district experiences violence be random, after conditioning on district and time fixed effects, and the occurrence of violence in the surrounding window of time (Equation (1)). Qualitatively, we find this assumption plausible because the precise timing (i.e., the day on which violence occurs) of insurgent attacks is often meant to surprise government forces. However, the assumption cannot be tested directly; we therefore perform several checks to assess if the occurrence of violence can be predicted beyond what our model in (1) captures. Specifically, we first regress the occurrence of violence on day t on the control variables in (1), i.e.,

$$\text{residuals } T_{dt} - \hat{E}(T_{dt} \mid \text{residuals})$$

). SI Appendix Table S1 assesses whether these can be predicted using recent lags and trends in the outcome variable (30-day displacement) and the number of subscribers observed to be in a district. We find that, using either a linear model or a machine learning approach (a random forest with 10-fold cross validation), these characteristics do not accurately predict residual violence ($R^2 \leq 0.000277$ (95% CI using non-parametric bootstrap, percentile interval: (0.00026, 0.0010)). Finally, as an additional robustness check, we find that adding more restrictive region*month time-varying fixed effects to model (1) does not qualitatively change the main results (SI Appendix Figure S5).

Comment 3:

The finding that individuals appear to anticipate violence is largely due to the fact that Afghan forces and the Taliban routinely warned civilians of major military operations (the kind most likely to be captured by the GED dataset). Civilians were often given a specified time to leave their villages before the operation began. As a result, the pre-violence migration observed in the findings is likely due less to civilian networks and other (unrecorded) violence than to efforts by the ANDSF and Taliban to warn villagers via leaflets, word of mouth, and SMS messages.

Thank you for this very helpful comment. We have added it to the *Discussion* as the first possible explanation of the anticipatory effect. The relevant section now reads:

Our analysis also indicates that people appear to anticipate the occurrence of violence, leaving before it occurs --- a finding that relates to prior work on the predictability of violence and conflict (Ward et al. 2010, Hegre et al. 2013, Cederman and Weidmann 2017, Bazzi et al. 2021, Kapoor and Narayanan 2021), including recent work using mobile phone data (Berger et al. 2014). This anticipatory effect is most pronounced with recently experienced violence (SI Appendix Figure S3). There are several possible related explanations for the anticipatory response we observe in Afghanistan. First, both NATO forces and the Taliban frequently warned civilians prior to major operations, for example by distributing leaflets (Baruch and Neuman 2011) and “night letters” (Johnson et al. 2017). Related, it may be that people were not anticipating a specific event, but were rather responding to a general period of unrest; individuals might perceive a threat of violence before a recorded event actually takes place. (Note that this does not violate the causal identification assumption of no unobserved time-varying confounders, since the perceived threat of violence does not cause violence to occur). For example, there might be skirmishes between armed forces that do not lead to fatalities or are not reported in the media.

Comment 4:

I'd note an additional explanation for why villagers headed to the provincial capitals: that's where Western and Afghan NGOs had the best distribution networks for humanitarian aid. In line with some of these findings, it's apparent that the effectiveness of aid after civilian casualties was sensitive to location (including proximity to bases and to larger urban centers; see Lyall 2019), so it makes sense that individuals head to these locations.

This is a great point. We have added this to the *Discussion* as a possible explanation: “The attraction of provincial capitals is likely due to several factors. First, capitals have a higher concentration of government security forces, and in the wake of violence, physical security is likely a crucial consideration. Related, humanitarian aid -- whose effectiveness depends on the security of the location (Lyall 2019) -- is often most easily accessed in provincial capitals.”

Comment 5:

The GED has a high degree of missingness for Afghanistan. Have the authors tried using ACLED as a robustness check? It typically has a higher degree of coverage for Afghanistan than Uppsala.

Thank you for this suggestion. We agree that this would be a useful exercise, unfortunately, to the best of our knowledge, ACLED data is not available for Afghanistan during the period of our study (2013-2017); coverage only begins in January 2017. This is true both for data available through the ACLED dashboard, as well as in the public documentation provided on the ACLED website.

Instead, we have expanded the section on violence data limitations to include a discussion of alternative sources of data on violence in Afghanistan, including ACLED, the Global Terrorism Database, and SIGACTS. The relevant paragraph now reads:

We also note limitations of the data we use to measure violence and conflict. We rely on data collected by the Uppsala Conflict Data Program (UCDP), which has known limitations that we discuss below. We also considered several alternative sources of data on violence, including the Armed Conflict Location and Event Data Project (ACLED) (Eck 2012), the Global Terrorism Database (GTD) (LaFree and Dugan 2007), and data on Significant Activities (SIGACTS) (Condra et al. 2018). However, these alternative data sources either do not have full data available during our period of study (ACLED and SIGACTS), or focus on specific types of violence (i.e., GTD focuses on terrorism). Thus, the UCDP data are most suitable for our purposes, but they have several limitations: they are collected from media sources, supplemented by NGO and IGO reports, field reports, and books (Sundberg and Melander 2013). However, not all events are included in media reports; inclusion is affected by a large number of factors (Öberg and Sollenberg 2011). For example, more populous regions and economic centers have a larger media presence and are thus better covered than peripheral areas; contested areas and places with less infrastructure tend to have poorer coverage. Media coverage is also influenced by how “newsworthy” the event is likely to be (Galtung and Ruge 1965). Our analysis is further restricted to those events for which the date and location (i.e., the district) are known. Since the spatial and temporal precision of event reporting is not random (Weidmann 2015), and that non-randomness may even differ across regions of Afghanistan, this could introduce bias into our analysis. For instance, if the threshold for reporting violence in rural areas is higher than the threshold in urban areas (Galtung and Ruge 1965, Öberg and Sollenberg 2011), that could bias our analysis to finding larger effects of violence in rural areas. We perform robustness tests in Materials and Methods (Section J) to assess the likely magnitude of this bias, but we also acknowledge that this is a limitation of our data -- and a limitation shared by most empirical work on conflict. While these problems of exclusion are faced by all

violent events data sets that rely on news sources, and there is little we can do to improve the quality of the data, we are optimistic that the quality of conflict event data will improve as they gain more widespread use. For instance, several papers now articulate best practices for the collection of conflict data (Salehyan 2015, Wang et al. 2016), and researchers are increasingly exploring how social media and other non-traditional data can be used to source conflict data (Zeitzoff 2011).

Comment 6:

One final note. I applaud the authors for being able to get these data (presumably from **[REDACTED]**, though the telecomm is never identified). I am concerned, however, that these data will not be made available for replication purposes to other scholars. A "small subset of the simulated data" is not especially helpful for replication. I'm not sure how to square this circle, but withholding the data seems at odds with our renewed emphasis on data transparency and availability.

We agree this is an important point. There are legal and ethical issues that prevent us from publicly releasing these data, and of course the current circumstances in Afghanistan only complicate matters. However, we do agree that replication is important, and so have updated our data availability statement to create additional options. In particular, we note that, with permission from the operator, we can provide district-level aggregate measures of migration. In conjunction with all of the code (that is publicly available on GitHub), this would allow for replication of all the analysis without the need for more sensitive phone records.

The data availability statement now reads:

With the exception of the mobile phone metadata, all data used in this paper are publicly available and sources have been listed in the text. The mobile phone dataset contains detailed information on roughly 20 billion mobile phone transactions in Afghanistan. These data contain proprietary and confidential information belonging to a private telecommunications operator, and cannot be publicly released. Upon reasonable request, we can provide information to accredited academic researchers about how to request the proprietary data from the telecommunications operator. With the telecommunication operator's permission, we can also provide district-level aggregate measures of migration for replication purposes to accredited academic researchers.

Decision Letter, third revision:

20th January 2022

Dear Josh,

Thank you for submitting your revised manuscript "Mobile Phone Data Reveal the Effects of Violence on Internal Displacement" (NATHUMBEHAV-210515394C). It has now been seen by one of the original referees and their comments are below. As you can see, the reviewer finds that the paper has improved in revision. We will therefore be happy in principle to publish it in *Nature Human Behaviour*, pending minor revisions to comply with our editorial and formatting guidelines.

We are now performing detailed checks on your paper and will send you a checklist detailing our editorial and formatting requirements within two weeks. Please do not upload the final materials and make any revisions until you receive this additional information from us.

Sincerely,
Aisha

Aisha Bradshaw, PhD
Senior Editor
Nature Human Behaviour

Reviewer #4 (Remarks to the Author):

Dear authors:

The authors should be commended for taking the extra steps to address my concerns about the as-if assumption and the local dynamics of displacement in Afghanistan. I would support publication of this manuscript.

Final Decision Letter:

Dear Josh,

We are pleased to inform you that your Article "Mobile Phone Data Reveal the Effects of Violence on Internal Displacement in Afghanistan", has now been accepted for publication in *Nature Human Behaviour*.

Please note that *Nature Human Behaviour* is a Transformative Journal (TJ). Authors whose manuscript was submitted on or after January 1st, 2021, may publish their research with us through the traditional subscription access route or make their paper immediately open access through payment of an article-processing charge (APC). Authors will not be required to make a final decision about access to their article until it has been accepted. IMPORTANT NOTE: Articles submitted before January 1st, 2021, are not eligible for Open Access publication. Find out more about Transformative Journals

With best regards,
Aisha

Aisha Bradshaw, PhD
Senior Editor
Nature Human Behaviour